# MESSY Estimation: Maximum-Entropy based Stochastic and Symbolic densitY Estimation

**Tony Tohme**[*]                                                   *tohme@mit.edu*
*Massachusetts Institute of Technology, USA.*

**Mohsen Sadr**[*]                                                   *msadr@mit.edu*
*Massachusetts Institute of Technology, USA.*
*Paul Scherrer Institute, Switzerland.*

**Kamal Youcef-Toumi**                                              *youcef@mit.edu*
*Massachusetts Institute of Technology, USA.*

**Nicolas G. Hadjiconstantinou**                                    *ngh@mit.edu*
*Massachusetts Institute of Technology, USA.*

**Reviewed on OpenReview:** *https://openreview.net/forum?id=Y2ruOLuQeS*

## Abstract

We introduce **MESSY** estimation, a **M**aximum-**E**ntropy based **S**tochastic and **S**ymbolic densit**Y** estimation method. The proposed approach recovers probability density functions symbolically from samples using moments of a Gradient flow in which the ansatz serves as the driving force. In particular, we construct a gradient-based drift-diffusion process that connects samples of the unknown distribution function to a guess symbolic expression. We then show that when the guess distribution has the maximum entropy form, the parameters of this distribution can be found efficiently by solving a linear system of equations constructed using the moments of the provided samples. Furthermore, we use Symbolic regression to explore the space of smooth functions and find optimal basis functions for the exponent of the maximum entropy functional leading to good conditioning. The cost of the proposed method for each set of selected basis functions is linear with the number of samples and quadratic with the number of basis functions. However, the underlying acceptance/rejection procedure for finding optimal and well-conditioned bases adds to the computational cost. We validate the proposed MESSY estimation method against other benchmark methods for the case of a bi-modal and a discontinuous density, as well as a density at the limit of physical realizability. We find that the addition of a symbolic search for basis functions improves the accuracy of the estimation at a reasonable additional computational cost. Our results suggest that the proposed method outperforms existing density recovery methods in the limit of a small to moderate number of samples by providing a low-bias and tractable symbolic description of the unknown density at a reasonable computational cost.

## 1 Introduction

Recovering probability density functions from samples is one of the fundamental problems in statistics with many applications. For example, the traditional task of discovering the underlying dynamics governing the corresponding distribution function is strongly dependent on the quality of the density estimator (Rudy et al., 2017). Applications include particle physics (Patrignani et al., 2016), boundary conditions for multi-scale kinetic problems (Frezzotti et al., 2005; Kon et al., 2014), and machine learning (Song et al., 2020).

---

[*]Equal contribution.

Broadly speaking, two categories of methods have been developed for this task: parametric and non-parametric estimators. While parametric methods assume a restrictive ansatz for the underlying distribution function, non-parametric methods provide a more flexible density estimate by performing a kernel integration locally using nearby samples. Although non-parametric methods do not need any prior knowledge of the underlying distribution, they suffer from the unclear choice of kernel and its support leading to bias and lack of moment matching. Examples of non-parametric density estimators include histogram and Kernel Density Estimation (KDE) (Rosenblatt, 1956; Jones et al., 1996; Sheather, 2004).

On the other hand, parametric density estimators may allow matching of moments while introducing modeling error, since a guess for the distribution is required. Parametric distributions include Gaussian, orthogonal expansion with respect to Gaussian using Hermite polynomials (also known as Grad's ansatz in kinetic theory) (Hermite, 1864; Grad, 1949; Cai et al., 2015), wavelet density estimation (Donoho et al., 1996), and Maximum Entropy Distribution (MED) (Kapur, 1989; Tagliani, 1999; Khinchin, 2013; Hauck et al., 2008) function among others. Given only the mean and variance, information theory provides us with the Gaussian distribution function as the least biased density, which has been used extensively in the literature. However, including higher order moments in a similar way, i.e. *moment problem*, raises further complications. For example in the context of kinetic theory, Grad proposed a closure that incorporates higher-order moments by considering a deviation from Gaussian using Hermite polynomials. Even though the information from higher moments is incorporated as the parameters of the polynomial expansion in Grad's ansatz, such a formulation suffers from not guaranteeing positivity of the estimated density along with the introduction of bias.

Among parametric density estimators, the Maximum Entropy Distribution (MED) function has been proposed in information theory as the least biased density estimate given a number of moments of the unknown distribution (Kapur, 1989). While MED provides the least biased density estimate, it suffers from two limitations. First, the distribution parameters (Lagrange multipliers) can only be found by solving a convex optimization problem with ill-conditioned Hessian (Dreyer, 1987; Levermore, 1996). The condition number increases either by increasing the order of the matching moments or approaching the limit of physical realizability which motivated the use of adaptive basis functions (Abramov, 2007; 2009). Second, MED only exists and is unique in bounded domains. While existence/uniqueness is guaranteed for recovering the distribution in the subspace occupied by the samples, the computational complexity associated with the direct computation of Lagrange multipliers has prevented researchers from deploying MED in practice.

**Related methods.** The problem of recovering a distribution function from samples has been investigated and studied before. We briefly review some of the work most relevant to our paper:

*Data-driven maximum entropy distribution function:* Several attempts have been made in the literature to speed up the computation of Lagrange multipliers for MED using Neural Networks (Sadr et al., 2021; Porteous et al., 2021; Schotthöfer et al., 2022) and Gaussian process regression (Sadr et al., 2020). Unfortunately, these approaches are data-dependent with support only on the trained subspace of distributions. Similar to the standard MED and other related closures, the data-driven MED can only handle polynomial moments as input, even though the data may be better represented with moments of other basis functions.

*Learning an invertible map:* The idea is to train an invertible neural network that maps the samples to a known distribution function. Then the unknown distribution function is found by inverting the trained map with the known distribution as the input. This procedure is called the normalizing flow technique (Rezende & Mohamed, 2015; Dinh et al., 2016; Kingma & Dhariwal, 2018; Durkan et al., 2019; Tzen & Raginsky, 2019; Kobyzev et al., 2020; Wang & Marzouk, 2022). This method has been used for re-sampling unknown distributions, e.g. Boltzmann generators (Noé et al., 2019), as well as density recovery such as AI-Feynmann (Udrescu & Tegmark, 2020; Udrescu et al., 2020). We note that AI-Feynman does not obtain the density from the samples directly; instead it first fits a density to the samples using the normalizing flow technique, constructs an input/output data set, then finds a simpler expression using symbolic regression. While invertible maps can be used to accurately predict densities, they can become expensive since for each problem one has to learn the parameters of the considered map via optimization.

*Diffusion map:* Instead of training for an invertible map, the diffusion map (Coifman et al., 2005; Coifman & Lafon, 2006) constructs coordinates using eigenfunctions of Markov matrices. Using pairwise distances between samples, in this method a kernel matrix is constructed as a generator of the underlying Langevin diffusion process. As shown by Li & Marzouk (2023), one can generate samples of the target distribution using Laplacian-adjusted Wasserstein gradient descent (Chewi et al., 2020). Unfortunately, this approach can become computationally expensive since it requires singular value decomposition of matrices of size equal to the number of samples.

*Gradient flow:* The gradient flow method has gained attention in recent years (Villani, 2009; Song et al., 2020; Song & Ermon, 2020). In particular, a class of sampling methods has been devised for drawing samples from a given distribution function using Langevin dynamics with the gradient of log-density as the driving force (Liu, 2017; Garbuno-Inigo et al., 2020a;b). Yet, this approach does not provide the density of the samples by itself. In our paper, we benefit from this formulation to recover the parameters of a density ansatz.

*Stein Variational Gradient Descent method (SVGD):* Given a target density function, the SVGD method as a deterministic and non-parameterized Gradient method generates samples of the target by carrying out a dynamic system on particles where the velocity field includes the grad-log of the target density (similar to the Gradient flow) and a kernel over particles (Liu & Wang, 2016). SVGD has been derived by approximating a kernelized Wasserstein gradient flow of KL divergence (Liu, 2017). While further steps have been taken to improve this method, e.g. SVGD with moment matching (Liu & Wang, 2018), similar to Gradient flow, this class of method cannot be used for estimating the density itself from samples.

*Wavelet and Conditional Renormalization Group method:* One of the powerful methods in signal processing is the wavelet method (Mallat, 1999), which may be considered as an extension of the Fourier method and domain decomposition. The basic idea is to consider the data on a multiple grids/scales, where the contribution from the smallest frequencies are found on the coarsest mesh and the highest frequencies on the finest mesh. While this method has been extended to finding high dimensional probability density functions (Marchand et al., 2022; Kadkhodaie et al., 2023), it is not clear how much bias is introduced by the orthogonal wavelet bases.

*KDE via diffusion:* In this method, the bandwidth of the kernel density estimation is computed using the minimum of mean integrated squared error and the fact that the KDE is the fundamental solution to a heat (more precisely Fokker-Planck) equation (Botev, 2007; Botev et al., 2010). While improvement has been achieved in this direction, we note that the KDE-diffusion method suffers from smoothing effects which introduce bias. Moreover moments of the unknown distribution are not necessarily matched.

*Symbolic regression:* Symbolic regression (SR) is a challenging task in machine learning that aims to identify analytical expressions that best describe the relationship between inputs and outputs of a given dataset. SR does not require any prior knowledge about the model structure. Traditional regression methods such as least squares (Wild & Seber, 1989), likelihood-based (Edwards, 1984; Pawitan, 2001), and Bayesian regression techniques (Lee, 1997; Leonard & Hsu, 2001; Tohme et al., 2020) use a fixed parametric model structure and only optimize for model parameters. SR optimizes for model structure and parameters simultaneously and hence is thought to be NP-hard, i.e. Non-deterministic Polynomial-time hard, (Udrescu & Tegmark, 2020; Petersen et al., 2021; Virgolin & Pissis, 2022). The SR problem has gained significant attention over recent years (Orzechowski et al., 2018; La Cava et al., 2021), and several approaches have been suggested in the literature. Most methods adopt genetic algorithms (Koza & Koza, 1992; Schmidt & Lipson, 2009; Tohme et al., 2023). Lately, researchers proposed using machine learning algorithms (e.g. Bayesian optimization, nonlinear least squares, neural networks, transformers, etc.) to solve the SR problem (Sahoo et al., 2018; Jin et al., 2019; Udrescu et al., 2020; Cranmer et al., 2020; Kommenda et al., 2020; Burlacu et al., 2020; Biggio et al., 2021; Mundhenk et al., 2021; Petersen et al., 2021; Valipour et al., 2021; Zhang et al., 2022; Kamienny et al., 2022). While most SR methods are concerned with finding a map from the input to the output, very few have addressed the problem of discovering probability density functions from samples (Udrescu et al., 2020).

**Our Contributions.** Our work improves the efficiency in determining the maximum entropy result for the unknown distribution. We specifically develop a new method for determining the unknown parameters (Lagrange multipliers) of this distribution without solving the optimization problem associated with this approach. This is achieved by relating the samples to the MED using Gradient flow, with the grad-log of the MED guess distribution serving as the drift. This results in a linear inverse problem for the Lagrange multipliers that is significantly easier to solve compared to the aforementioned optimization problem. We also propose a Monte Carlo search in the space of smooth functions for finding an optimal basis function for describing (the exponent of) the maximum entropy ansatz. As a selection criterion, we rate randomly created basis functions according to the condition number associated with the coefficient (Hessian) matrix of the inverse problem for the Lagrange multipliers. This helps to maintain good conditioning, which allows us to incorporate more degrees of freedom and recover the unknown density accurately. Discontinuous density functions are treated by considering only the domain supported by data and using a multi-level solution process.

The paper is organized as follows. In Section 2 we review the concept of Gradient flow with grad-log of a known density as the drift. In Section 3, we show how parameters of a guess MED may be found by computing the relaxation rates of the corresponding Gradient flow. Using the maximum entropy ansatz, in Section 4 we derive a linear inverse problem for finding the Lagrange multipliers without the need for solving an optimization problem. In Section 5, we propose a symbolic regression method for finding basis functions that can be used to increase degrees of freedom while maintaining good conditioning of the problem by construction. In Section 6, we propose a generalization of the maximum entropy ansatz that allows including further degrees of freedom in a multi-level fashion. Section 7 presents the complete MESSY algorithm. In Section 8, we validate MESSY by comparing its predictions to those of benchmark density estimators in recovering distributions featuring discontinuities and bi-modality, as well as distributions close to the limit of realizability. Finally, in Section 9, we offer our conclusions and outlook.

## 2 Gradient flow and theoretical motivation

Consider a set of samples of a random variable $\boldsymbol{X}$ from an unknown density distribution function $f(\boldsymbol{x})$. Let our guess for this distribution function, the "ansatz", be denoted by $\hat{f}(\boldsymbol{x})$.

Instead of constructing a non-parametric approximation of the target density numerically from samples of $\boldsymbol{X}$ (like histogram or KDE) and then calculating its difference from the guess density $\hat{f}$, in this work we suggest measuring the distance using transport. In particular, we use the fact that the steady-state distribution of $\boldsymbol{X}(t)$ which follows the stochastic differential equation (SDE)

$$d\boldsymbol{X} = \nabla_{\boldsymbol{x}}\big[\log\big(\hat{f}\big)\big]dt + \sqrt{2}d\boldsymbol{W}_t \tag{1}$$

is the distribution $\hat{f}$. Here, $\boldsymbol{W}_t$ is the standard Wiener process of dimension $\dim(\boldsymbol{x})$. We note that Eq. (1) is known as the gradient flow (or Langevin dynamics) with grad-log of density as the force. We note that this drift differs from the score-based generative model in Song et al. (2020) where the drift is a function of time, i.e. $\nabla_{\boldsymbol{x}}\big[\log(\hat{f}(t))\big]$.

The distance of $f$ from $\hat{f}$ may be measured by the time required for the SDE with $\boldsymbol{X}(t=0) \sim f$ to reach steady state. Alternatively, one may compare the moments computed from the solution to $\boldsymbol{X}(t)$ against the input samples to measure this distance. Both these approaches are subject to numerical and statistical noise associated with the numerical scheme deployed in integrating Eq. (1). In the next section, we derive an efficient way of computing the parameters of our approximation $\hat{f}$ based on these ideas. We also show that the transition from $f$ to $\hat{f}$ is monotonic.

## 3 Ansatz as the target density of Gradient flow

According to Ito's lemma (Platen & Bruti-Liberati, 2010) the transition of $f$ to $\hat{f}$ is governed by the Fokker-Planck equation

$$\frac{\partial f}{\partial t} = \nabla_{\boldsymbol{x}} \left[ \hat{f} \nabla_{\boldsymbol{x}} [f/\hat{f}] \right] \tag{2}$$

$$= -\nabla_{\boldsymbol{x}} \cdot \left[ \nabla_{\boldsymbol{x}} [\log(\hat{f})] f \right] + \nabla_{\boldsymbol{x}}^2 [f] . \tag{3}$$

**Proposition 3.1.** *The distribution function $f(t)$ governed by the Fokker-Planck Eq. (2) converges to $\hat{f}$ as $t \to \infty$. Furthermore, the cross entropy distance between $f$ and $\hat{f}$ monotonically decreases during this transition.*

*Proof.* Let us multiply both sides of Eq. (2) by $\log(f/\hat{f})$ and take the integral with respect to $\boldsymbol{x}$ in order to obtain the evolution of the cross-entropy $S = \int f \log(f/\hat{f}) d\boldsymbol{x}$. It follows that

$$\frac{dS}{dt} = \int \log(f/\hat{f}) \nabla_{\boldsymbol{x}} \left[ \hat{f} \nabla_{\boldsymbol{x}} [f/\hat{f}] \right] d\boldsymbol{x}$$

$$= \int \nabla_{\boldsymbol{x}} \left[ \hat{f} \log(f/\hat{f}) \nabla_{\boldsymbol{x}} [f/\hat{f}] \right] d\boldsymbol{x} - \int \hat{f} \nabla_{\boldsymbol{x}} [\log(f/\hat{f})] \cdot \nabla_{\boldsymbol{x}} [f/\hat{f}] d\boldsymbol{x}$$

$$= \underbrace{\int \nabla_{\boldsymbol{x}} \left[ \hat{f} \log(f/\hat{f}) \frac{f}{\hat{f}} \nabla_{\boldsymbol{x}} [\log(f/\hat{f})] \right] d\boldsymbol{x}}_{=0} - \int \hat{f} \nabla_{\boldsymbol{x}} [\log(f/\hat{f})] \cdot \frac{f}{\hat{f}} \nabla_{\boldsymbol{x}} [\log(f/\hat{f})] d\boldsymbol{x}$$

$$= -\sum_{i=1}^{\dim(\boldsymbol{x})} \int f \left( \nabla_{x_i} [\log(f/\hat{f})] \right)^2 d\boldsymbol{x} \le 0 . \tag{4}$$

Here, we use the regularity condition that $f \log(f/\hat{f}) \nabla_{\boldsymbol{x}} \log(f/\hat{f}) \to 0$ as $|\boldsymbol{x}| \to \infty$. Therefore, given any initial condition for $f$ at $t = 0$, the cross-entropy distance between $f$ and $\hat{f}$ following the Fokker-Planck in Eq. (2) monotonically decreases until it reaches the steady-state with the trivial fixed point $f \to \hat{f}$ as $t \to \infty$. For details, see (Liu, 2017). $\qquad\square$

Instead of seeking solutions of Eq. (2), our approach focuses on working with appropriate empirical moments of this equation, which can be evaluated from the available samples. As will be demonstrated below, this approach lends itself to a very effective method for determining $\hat{f}$.

Let us denote a vector of basis functions in $\mathbb{R}^{\dim(\boldsymbol{x})}$ by $\boldsymbol{H}(\boldsymbol{x})$. By multiplying both sides of Eq. (3) by $\boldsymbol{H}(\boldsymbol{x})$ and integrating with respect to $\boldsymbol{x}$, we obtain the evolution equation for the moments, also known as the relaxation rates,

$$\frac{d}{dt} \left[ \int \boldsymbol{H} f d\boldsymbol{x} \right] = -\int \boldsymbol{H} \nabla_{\boldsymbol{x}} \cdot \left[ \nabla_{\boldsymbol{x}} [\log(\hat{f})] f \right] d\boldsymbol{x} + \int \boldsymbol{H} \nabla_{\boldsymbol{x}}^2 [f] d\boldsymbol{x} . \tag{5}$$

Assuming that the underlying density $f$ is integrable in $\mathbb{R}^{\dim(\boldsymbol{x})}$ and $f\boldsymbol{H} \to \boldsymbol{0}$ as $\boldsymbol{x} \to \infty$, which is implied by the existence of moments, we use integration by parts to obtain

$$\frac{d}{dt} \left[ \int \boldsymbol{H} f d\boldsymbol{x} \right] = \int \nabla_{\boldsymbol{x}} [\boldsymbol{H}] \cdot \nabla_{\boldsymbol{x}} [\log(\hat{f})] f d\boldsymbol{x} + \int \nabla_{\boldsymbol{x}}^2 [\boldsymbol{H}] f d\boldsymbol{x} . \tag{6}$$

Given samples of $f$, one can compute the relaxation rates of moments represented by Eq. (6) as a measure of the difference between $\hat{f}$ and $f$. These relaxation rates can be used as the gradient in the search for parameters of a given ansatz, i.e.

$$\boldsymbol{g}(t) = \frac{d}{dt} \left\langle \boldsymbol{H}(\boldsymbol{X}(t)) \right\rangle = \left\langle \nabla_{\boldsymbol{x}} [\boldsymbol{H}(\boldsymbol{X}(t))] \cdot \nabla_{\boldsymbol{x}} [\log(\hat{f}(\boldsymbol{X}(t)))] \right\rangle + \left\langle \nabla_{\boldsymbol{x}}^2 [\boldsymbol{H}(\boldsymbol{X}(t))] \right\rangle . \tag{7}$$

In the above, $\langle \phi(\boldsymbol{X}) \rangle$ denotes the unbiased empirical measure for the expectation of $\phi(\boldsymbol{X})$ which is computed using samples of $\boldsymbol{X}_i$, for $i = 1, ..., N$ via $\langle \phi(\boldsymbol{X}) \rangle = \frac{1}{N} \sum_{i=1}^{N} \phi(\boldsymbol{X}_i)$.

In what follows we develop an approach that uses $g(t)$ as the gradient of an optimization problem to bring computational benefits to the solution of the maximum entropy problem.

## 4 Maximum Entropy Distribution as an ansatz for the gradient flow

In this work, we use the maximum entropy distribution function as our parameterized ansatz for $\hat{f}$, i.e.

$$\hat{f}(\boldsymbol{x}) = Z^{-1} \exp\left(\boldsymbol{\lambda} \cdot \boldsymbol{H}(\boldsymbol{x})\right) \tag{8}$$

where $Z = \int \exp(\boldsymbol{\lambda} \cdot \boldsymbol{H}(\boldsymbol{x})) d\boldsymbol{x}$ is the normalization constant. The motivation for choosing this family of distributions is the fact that this is the least-biased distribution for the moment problem, provided the given moments are matched.

**Definition 4.1.** *Moment problem*

*The problem of finding a distribution function $f(\boldsymbol{x})$ given its moments $\int \boldsymbol{H}(\boldsymbol{x}) f(\boldsymbol{x}) d\boldsymbol{x} = \boldsymbol{\mu}$ for the vector of basis functions $\boldsymbol{H}(\boldsymbol{x})$ will be referred to as the moment problem.*

In particular, the density in Eq. (8) is the extremum of the loss functional that minimizes the Shannon entropy with constraints on moments $\boldsymbol{\mu}$ using the method of Lagrange multipliers, i.e.

$$\hat{f}(\boldsymbol{x}) = \underset{\mathcal{F} \in \mathcal{K}}{\arg\min} \ \mathcal{C}[\mathcal{F}(\boldsymbol{x})] \tag{9}$$

$$\text{where} \quad \mathcal{C}[\mathcal{F}(\boldsymbol{x})] := \int \mathcal{F}(\boldsymbol{x}) \log(\mathcal{F}(\boldsymbol{x})) d\boldsymbol{x} - \sum_{i=1}^{N_b} \lambda_i \left( \int H_i(\boldsymbol{x}) \mathcal{F}(\boldsymbol{x}) d\boldsymbol{x} - \mu_i(\boldsymbol{x}) \right) . \tag{10}$$

Here $\mathcal{K}$ denotes the space of probability density functions with measurable moments; see (Kapur, 1989) and Appendix A for more details. In this paper, we denote the number of considered basis functions by $N_b$, while $N_m$ denotes the highest order of these basis functions. For instance, in the case of traditional one-dimensional random variable where polynomial basis functions are deployed, i.e. $\boldsymbol{H} = \left[ x, x^2, ..., x^{N_m} \right]$, we have $N_m = N_b$.

Here, we use the following definition for the growth rate of a basis function.

**Definition 4.2.** *Growth rate of n-th order*

*A function $\psi(x)$ has the growth-rate of n-th order if $|\psi(x)| \le Cx^n$ for all $x \ge x_0$ where $C \in \mathbb{R}^+$ and $x_0 \in \mathbb{R}$. This is often denoted by $\psi(x) = \mathcal{O}(x^n)$.*

We note that the moment problem for the MED is reduced to the following optimization problem by substituting the extremum 8 back in the objective functional equation 10, see e.g. Abramov (2006).

**Definition 4.3.** *Standard dual optimization problem of Maximum entropy distribution function*

Given moments $\boldsymbol{\mu}$, the Lagrange multipliers $\boldsymbol{\lambda}$ of the maximum entropy distribution are the solution to the following unconstrained optimization problem

$$\boldsymbol{\lambda} = \underset{\hat{\boldsymbol{\lambda}} \in \mathbb{R}_b^N}{\arg\min} \left\{ \log \left[ \int \exp(\hat{\boldsymbol{\lambda}} \cdot \boldsymbol{H}(\boldsymbol{x})) d\boldsymbol{x} \right] - \hat{\boldsymbol{\lambda}} \cdot \boldsymbol{\mu} \right\} . \tag{11}$$

Clearly, the standard optimization problem for finding Lagrange multipliers is nonlinear and requires deploying iterative methods, such as the Newton-Raphson method detailed in A.

Instead, using the Gradient flow, we find an alternative optimization problem for finding Lagrange multipliers which is linear and simple to compute from given samples.

Substituting Eq. (8) for $\hat{f}$ in Eq. (7) results in the relaxation rate

$$\boldsymbol{g}(t) = \sum_{i=1}^{\dim(\boldsymbol{x})} \left\langle \nabla_{x_i}[\boldsymbol{H}(\boldsymbol{X}(t))] \otimes \nabla_{x_i}[\boldsymbol{H}(\boldsymbol{X}(t))] \right\rangle \boldsymbol{\lambda} + \sum_{i=1}^{\dim(\boldsymbol{x})} \left\langle \nabla_{x_i}^2[\boldsymbol{H}(\boldsymbol{X}(t))] \right\rangle , \tag{12}$$

where $\otimes$ indicates the outer product. Let us define the matrix $\boldsymbol{L}^{\mathrm{ME}}$ as

$$\boldsymbol{L}^{\mathrm{ME}}(t) := \sum_{i=1}^{\dim(\boldsymbol{x})} \left\langle \nabla_{x_i}[\boldsymbol{H}(\boldsymbol{X}(t))] \otimes \nabla_{x_i}[\boldsymbol{H}(\boldsymbol{X}(t))] \right\rangle . \tag{13}$$

**Definition 4.4.** *Optimization problem of Maximum entropy distribution function via Gradient flow*

Given samples $\boldsymbol{X}$ of the target distribution $f$, Lagrange multipliers $\boldsymbol{\lambda}$ of maximum entropy distribution estimate $\hat{f}$ is the solution to the following unconstrained optimization problem

$$\boldsymbol{\lambda} = \underset{\hat{\boldsymbol{\lambda}} \in \mathbb{R}_b^N}{\arg\min} \left\{ \boldsymbol{L}^{\mathrm{ME}}(t) : \frac{\hat{\boldsymbol{\lambda}} \otimes \hat{\boldsymbol{\lambda}}}{2} + \sum_{i=1}^{\dim(\boldsymbol{x})} \left\langle \nabla_{x_i}^2[\boldsymbol{H}(\boldsymbol{X}(t))] \right\rangle \cdot \hat{\boldsymbol{\lambda}} \right\} , \tag{14}$$

where (:) denotes the Frobenius inner (or double dot) product, i.e. $\boldsymbol{A} : \boldsymbol{B} = \sum_{i,j} A_{ij} B_{ij}$ for given two-dimensional tensors $\boldsymbol{A}$ and $\boldsymbol{B}$.

We note that the matrix $\boldsymbol{L}^{\mathrm{ME}}$ is the Hessian of the optimization problem with gradient given by Eq. (12) which is positive definite, making the underlying optimization problem convex.

**Proposition 4.5.** *The Hessian matrix $\boldsymbol{L}^{\mathrm{ME}}$ is symmetric positive definite. As a result, the optimization problem with gradient given by Eq. (12) and Hessian matrix given by Eq. (13) is strictly convex.*

*Proof.* Clearly, the Hessian matrix defined by Eq. (13) is symmetric, i.e. $L_{i,j}^{\mathrm{ME}} = L_{j,i}^{\mathrm{ME}} \ \forall i,j = 1, ..., N_b$. We further note that this matrix is positive definite, i.e. for any non-zero vector $\boldsymbol{w} \in \mathbb{R}^{N_b}$ we can write

$$\boldsymbol{w}^T \boldsymbol{L}^{\mathrm{ME}}(t) \boldsymbol{w} = \sum_{i=1}^{\dim(\boldsymbol{x})} \left\langle \boldsymbol{w}^T \nabla_{x_i}[\boldsymbol{H}(\boldsymbol{X}(t))] \ \nabla_{x_i}[\boldsymbol{H}(\boldsymbol{X}(t))]^T \boldsymbol{w} \right\rangle \tag{15}$$

$$= \sum_{i=1}^{\dim(\boldsymbol{x})} \left\langle \left( \boldsymbol{w}^T \nabla_{x_i}[\boldsymbol{H}(\boldsymbol{X}(t))] \right)^2 \right\rangle > 0 . \tag{16}$$

Given the Hessian is symmetric positive definite, we conclude that the underlying optimization problem is convex (Chong & Zak, 2013). $\square$

When the matrix $\boldsymbol{L}^{\mathrm{ME}}$ is well-conditioned, we can directly compute the Lagrange multipliers using samples, i.e. a linear solution to the optimization problem in Def. 4.4. This can be achieved by solving Eq. (12) for the Lagrange multipliers

$$\boldsymbol{L}^{\mathrm{ME}}(t) \boldsymbol{\lambda} = \boldsymbol{g}(t) - \sum_{i=1}^{\dim(\boldsymbol{x})} \left\langle \nabla_{x_i}^2[\boldsymbol{H}(\boldsymbol{X}(t))] \right\rangle \tag{17}$$

for a given relaxation rate $\boldsymbol{g}$.

We proceed by noting that a convenient way for determining the parameters of $\hat{f}$ is to set $\hat{f} = f(t = 0)$ in the above formulation, or in other words, require that the given samples are also samples of $\hat{f}$ as given. This corresponds to the steady solution of Eq. (12), namely $\boldsymbol{g} \to \boldsymbol{0}$, which implies the remarkably simple result

$$\boldsymbol{\lambda} = -\left( \boldsymbol{L}^{\mathrm{ME}} \right)^{-1} \left( \sum_{i=1}^{\dim(\boldsymbol{x})} \left\langle \nabla_{x_i}^2[\boldsymbol{H}(\boldsymbol{X}(t = 0))] \right\rangle \right) , \tag{18}$$

which implies a closed-form solution for the Lagrange multipliers through the above linear problem.

While Eq. (18) analytically recovers the Lagrange multipliers $\boldsymbol{\lambda}$ directly from samples of $\boldsymbol{X}$, it still requires inverting the matrix $\boldsymbol{L}^{\mathrm{ME}}$ which may be ill-conditioned (Abramov, 2010; Alldredge et al., 2014). This means that the resulting Lagrange multipliers may become sensitive to noise in the samples and the choice of the basis functions. In order to cope with this issue, we propose computing $\boldsymbol{\lambda}$ as outlined below.

**Orthonormalizing the basis functions.** We construct an orthonormal basis function with respect to $\boldsymbol{X} \sim f$ using the modified Gram-Schmidt algorithm as described in Algorithm 1. We deploy the orthonormal basis functions from the Gram-Schmidt procedure to construct $\nabla_{\boldsymbol{x}}[\boldsymbol{H}]^{\perp}$, i.e. $\nabla_{\boldsymbol{x}}[\boldsymbol{H}]$ is the input to Algorithm 1, and by integration we obtain $\boldsymbol{H}^{\perp}$. This leads to a well-conditioned matrix $\boldsymbol{L}^{\mathrm{ME}}$, since the resulting matrix should be close to identity $\boldsymbol{L}^{\mathrm{ME}} \approx \boldsymbol{I}$ with condition number $\mathrm{cond}\big(\boldsymbol{L}^{\mathrm{ME}}\big) \approx 1$ subject to round-off error. We note that the cost of this algorithm is quadratic with the number of basis functions and linear with the number of samples.

---

**Algorithm 1:** Modified Gram-Schmidt: Given a vector of basis functions $\boldsymbol{\phi}$, this algorithm constructs an orthonormal basis functions $\boldsymbol{\phi}^{\perp}$ with respect to $f$ such that $\langle \boldsymbol{\phi}^{\perp}(\boldsymbol{X}) \otimes \boldsymbol{\phi}^{\perp}(\boldsymbol{X}) \rangle \approx \boldsymbol{I}$ using the modified Gram-Schmidt procedure (Giraud et al., 2002; Abramov, 2010).

---

**Input:** $\boldsymbol{\phi}$
Initialize $\boldsymbol{\phi}^{\perp} \leftarrow \boldsymbol{\phi}$;
**for** $i = 1, ..., \dim(\boldsymbol{\phi})$ **do**
    $\phi_i^{\perp} = \phi_i^{\perp}/\sqrt{\langle(\phi_i^{\perp}(\boldsymbol{X}))^2\rangle}$;
    **for** $j = i + 1, ..., \dim(\boldsymbol{\phi})$ **do**
        $\phi_j^{\perp} \leftarrow \phi_j^{\perp} - \langle\phi_i^{\perp}(\boldsymbol{X})\phi_j^{\perp}(\boldsymbol{X})\rangle\phi_i^{\perp}$;
    **end**
**end**
**Return** $\boldsymbol{\phi}^{\perp}$

---

### 4.1 Comparing the proposed formulation to standard Maximum Entropy Distribution

Here we point out several advantages of using the proposed loss function compared to the standard maximum entropy closure.

- **A closed-form solution.** By setting the relaxation rate of the moments to zero, the Lagrange multipliers can be computed directly from samples $\boldsymbol{X} \sim f$, i.e. by solving the system 18, without the need for the line-search associated with the Newton method of solving the optimization problem of standard MED, i.e. Def. 4.3. This is a significant improvement compared to standard MED, where Lagrange multipliers are estimated iteratively — see Eq. (A.5) and Algorithm 4 in Appendix A.

- **Avoiding the curse of dimensionality in integration.** In the proposed method, the computational complexity associated with the integration only depends on the number of samples, and not on the dimension of the probability space. The proposed method takes full advantage of having access to the samples of the unknown distribution function. This is in contrast to the standard Newton-Raphson method of solving optimization problem 4.3 where samples of the initial guessed MED corresponding to the initial guessed $\boldsymbol{\lambda}$ is not available and one runs to the curse of dimensionality in computation of gradient and Hessian.

  In particular, we compute the orthonormal basis function, gradient, and Hessian using the samples of $\boldsymbol{X}$. This use of the Monte Carlo integration method avoids the curse of high dimensionality associated with the conventional method for computing Lagrange multipliers. By deploying the Law of Large Numbers (LLN) in computing integrals, the proposed method benefits from the well-known result that the error of Monte Carlo integration is independent of dimension. In particular, given $N$ independent, identically distributed $d$-dimensional random variables $\boldsymbol{X}_1, ..., \boldsymbol{X}_N$ with probability density $f(\boldsymbol{x})$, where $\boldsymbol{X} \sim f$, the variance of the empirical estimator of a moment

$\phi(\boldsymbol{x})$ of $f$, i.e. $\mathbb{E}^{\Delta}[\phi(\boldsymbol{X})] = \sum_{i=1}^{N} \phi(\boldsymbol{X}_i)/N$, is

$$\text{Var}(\mathbb{E}^{\Delta}[\phi(\boldsymbol{X})]) = \frac{\text{Var}(\phi(\boldsymbol{x}))}{N} \tag{19}$$

which is independent of the dimension $d = \dim(\boldsymbol{x})$. Therefore, for a required ratio of variance in prediction and variance of the underlying random variable, i.e. $\text{Var}(\mathbb{E}^{\Delta}[\phi(\boldsymbol{X})])/\text{Var}(\phi(\boldsymbol{x}))$, the cost of integration is $\mathcal{O}(sN)$. The factor $s$ denotes the cost of computing $\phi(\boldsymbol{X})$ for one sample.

This is a considerable advantage compared to the standard approach of finding the Lagrange multipliers of MED where the cost associated with integration is of order $\mathcal{O}(N^d)$ where $N$ is the number of discretization points in each dimension. We remind the reader that in the standard Newton-Raphson method of finding the Lagrange multipliers, one updates the guessed $\boldsymbol{\lambda}$ by

$$\boldsymbol{\lambda} \leftarrow \boldsymbol{\lambda} - \boldsymbol{L}^{-1}(\boldsymbol{\lambda})\boldsymbol{g}(\boldsymbol{\lambda}) \tag{20}$$

where the gradient $\boldsymbol{g}$ and Hessian $\boldsymbol{L}$ need to be computed during each iteration, see section A and reference therein for details. Since samples of the guessed distribution, i.e. guessed $\boldsymbol{\lambda}$, are not available, computation of the gradient and Hessian can become expensive. Specifically, one has to either generate samples of the guess distribution in each intermediate step of the Newton-Raphson method which adds complexity, or deploy a deterministic integration method with cost $\mathcal{O}(N^d)$ where $N$ is the number of discretization points in each dimension and $d = \dim(\boldsymbol{x})$.

We further point out that since in both the proposed solution and the standard iterative method a matrix needs to be inverted, the cost in both algorithms scales $\mathcal{O}(N_b^3)$ with number of basis functions (moments) regardless of the cost associated with integration. For example, in case of using monomials up to 2nd order in all dimensions as basis functions i.e. $\boldsymbol{H} = [x_1, ..., x_d, x_1^2, x_1 x_2, ..., x_d^2]$, there are $|\boldsymbol{H}| = d + d(d+1)/2$ unique bases, leading to complexity $\mathcal{O}\left((d + d(d+1)/2)^3\right)$ for solving the linear system 18.

- **Relaxed existence requirements.** Since the proposed approach directly finds the Lagrange multipliers for the realizable moment problem linearly using Eq. (18), it avoids the problem of possible non-realizable distribution estimate with intermediate Lagrange multipliers that is present in the iterative methods. This is another advantage compared to the standard MED optimization problem, Eq. (4.3), where the line search Eq. (20) may fail as the distribution associated with the intermediate $\boldsymbol{\lambda}$ may not exist (not integrable). This is a common problem when finding Lagrange multipliers for the moment problem close to the limit of realizability when the condition number of the Hessian becomes large and the iterative Newton-Raphson method fails, e.g. see Alldredge et al. (2014).

## 5 Symbolic-Based Maximum Entropy Distribution

In the standard moment problem it is common to consider polynomials for the moment functions in $\boldsymbol{H}$, i.e. $\boldsymbol{H} = [x, x^2, \ldots]$, even though other basis functions may better represent the unknown distribution. Additionally, such polynomial basis functions are notorious for resulting in ill-conditioned solution processes. For these reasons, we introduce a symbolic regression approach to introduce some diversity and ultimately optimize over our use of basis functions. As we will see in the next section, adding the symbolic search to our MED description improves the accuracy, convergence, and robustness of the density recovery problem.

Before diving into the proposed method, we first briefly review the general task of symbolic regression.

**Definition 5.1.** *Symbolic Regression (SR) problem*

*Given a metric $\mathcal{L}$ and a dataset $\mathcal{D} = \{\boldsymbol{x}_i, y_i\}_{i=1}^{N}$ consisting of $N$ independent identically distributed (i.i.d.) paired samples, where $\boldsymbol{x}_i \in \mathbb{R}^{\dim(\boldsymbol{x})}$ and $y_i \in \mathbb{R}$, the SR problem searches in the space of functions $\mathcal{S}$ defined by a set of given mathematical functions (e.g., cos, sin, exp, ln) and arithmetic operations (e.g., $+$, $-$, $\times$, $\div$), for a function $\psi^*(\boldsymbol{x})$ which minimizes $\sum_{i=1}^{N} \mathcal{L}(y_i, \psi(\boldsymbol{x}_i))$ where $\psi \in \mathcal{S}$.*

In order to deploy the SR method for the density recovery, we need to restrict the space of functions $\mathcal{S}$ to those which satisfy non-negativity, normalization and existence of moments with respect to the vector of linearly independent (polynomial) basis functions $\boldsymbol{R}$. The space of such distributions can be defined as

$$\mathcal{S}_{f|\boldsymbol{R}} := \left\{ f(\boldsymbol{x}) \in \mathcal{S} \,\middle|\, f(\boldsymbol{x}) \geq 0 \;\forall \boldsymbol{x} \in \mathbb{R}^{\dim(\boldsymbol{x})}, \int_{\mathbb{R}^{\dim(\boldsymbol{x})}} f(\boldsymbol{x}) \, d\boldsymbol{x} = 1, \int_{\mathbb{R}^{\dim(\boldsymbol{x})}} \boldsymbol{R}(\boldsymbol{x}) f(\boldsymbol{x}) \, d\boldsymbol{x} < +\infty \right\} . \tag{21}$$

In order to ensure non-negativity, motivated by the MED formulation, we consider $\hat{f}$ to be exponential, i.e.

$$\hat{f}(\boldsymbol{x}) \propto \exp\left(\mathcal{G}(\boldsymbol{x})\right) \qquad \Longleftrightarrow \qquad \log\left(\hat{f}(\boldsymbol{x})\right) \propto \mathcal{G}(\boldsymbol{x}) , \tag{22}$$

where $\mathcal{G}(\boldsymbol{x})$ is an analytical (or symbolic) function of $\boldsymbol{x} = \left[x_1, x_2, \dots, x_{\dim(\boldsymbol{x})}\right]$. While the non-negativity is guaranteed, existence of moments needs to be verified when a test function for $\mathcal{G}(\boldsymbol{x})$ is considered. As our focus in this paper is on the maximum entropy distribution function given by Eq. (8), we consider $\mathcal{G}(\boldsymbol{x})$ to have the form

$$\mathcal{G}(\boldsymbol{x}) = \boldsymbol{\lambda} \cdot \boldsymbol{H}(\boldsymbol{x}) = \sum_{i=1}^{N_b} \lambda_i H_i(\boldsymbol{x}) . \tag{23}$$

Now we proceed to provide a modified formulation for SR tailored to our MED problem.

**Definition 5.2.** *Symbolic Regression for the Maximum Entropy Distribution (SR-MED) problem*

*Given a measure of difference between distributions $\mathcal{L}$ (e.g. KL Divergence) and a dataset $\mathcal{D} = \{\boldsymbol{X}_i\}_{i=1}^N$ consisting of $N$ i.i.d. samples, where $\boldsymbol{X}_i \in \mathbb{R}^{\dim(\boldsymbol{x})}$, the SR-MED problem searches in the space $\mathcal{S}^{N_b}$ for $N_b$ basis functions subject to $\hat{f} \in \mathcal{S}_{f|\boldsymbol{R}}$ which minimizes $\mathcal{L}$.*

Here, we deploy continuous functions consisting of *binary* operators (e.g. $+$, $-$, $\times$, $\div$) or *unary* functions (e.g. cos, sin, exp, log) to fill the space $S^{N_b}$. As in most of the SR methods, we encode mathematical expressions using symbolic expression trees, a type of binary tree, where internal nodes contain operators or functions and terminal nodes (or leaves) contain input variables of constants. For instance, the expression tree in Figure 1 represents $x^2 \cos(x)$. In this paper, we perform a Monte Carlo symbolic search in the space of smooth functions (by generating random expression trees) to find a vector of basis functions $\boldsymbol{H}$ that guarantees acceptable $\text{cond}(\boldsymbol{L}^{\text{ME}})$, by rejecting candidates that do not satisfy this condition. In our search, we do not consider test basis functions with odd growth rates which lead to non-realizable distributions.

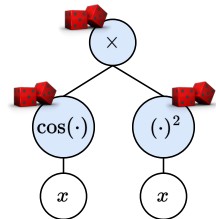

Figure 1: Expression tree for $x^2 \times \cos(x)$.

## 6 Multi-level density recovery

We further improve our proposed method by introducing a multi-level process that improves our prediction as the distribution becomes more detailed. The goal is to obtain a more generalized MED estimate with the form

$$\hat{f}(\boldsymbol{x}) = \sum_{l=1}^{N_L} m^{[l]} \, \hat{f}^{[l]}(\boldsymbol{x}) \tag{24}$$

$$\text{where} \quad \hat{f}^{[l]}(\boldsymbol{x}) = \frac{1}{Z^{[l]}} \exp\left(\boldsymbol{\lambda}^{[l]} \cdot \boldsymbol{H}^{[l]}(\boldsymbol{x})\right), \tag{25}$$

$(.)^{[l]}$ denotes the level index, $Z^{[l]}$ is the normalization factor of density at level $l$, $N_L$ is the number of levels considered and $m^{[l]}$ indicates the portion of total mass that is covered by $\hat{f}^{[l]}$. We note that this multi-level approach is recursive and can be described as follows:

- **Step 1: Find MED estimate $\hat{f}^{[l]}$ at level $l$.** At level $l$, first we pick a basis function $\boldsymbol{H}^{[l]}$ by solving the SR-MED problem detailed in Def. 5.2. Then, we orthonormalize the basis function with respect to the distribution of the samples using Gram–Schmidt's procedure as outlined in Algorithm 1.

- **Step 2: Removing subset of samples covered by $\hat{f}^{[l]}$.** Here, we attempt to find and remove a subset of samples $\mathcal{D}_{\text{mask}}^{[l]}$ – representing a fraction of the *mass*, i.e. $m^{[l]} = |\mathcal{D}_{\text{mask}}^{[l]}|/|\mathcal{D}|$ – that can be estimated by our estimated $\hat{f}^{[l]}$ at this level. To this end, we deploy acceptance/rejection with probability $\hat{f}^{[l]}/\hat{f}^{\text{hist}}$ to find and remove $\mathcal{D}_{\text{mask}}^{[l]}$ from the remaining samples $\mathcal{D}^{[l]}$.

- **Step 3: Repeat steps 1-2 for the next level $l+1$ until almost no samples are left.** Repeat steps 1-2 with the remaining uncovered samples (which constitutes the next level) until there are (almost) no uncovered samples. The resulting total distribution is a weighted sum of the estimates from each level.

In Algorithm 2, we detail a pseudocode for our devised multi-level process. As we will see in the next section, our proposed multi-level recursive mechanism improves overall performance, and elegantly describe details of multi-mode distributions.

---

**Algorithm 2:** Multi-level, symbolic and recursive algorithm for density recovery. Here, $\mathcal{D}^{[l]}$ denotes the set of samples at level $l$ and $\boldsymbol{u}$ is a random variable that is uniformly distributed in $(0,1)$, i.e. $\boldsymbol{u} \sim \mathcal{U}([0,1])$.

---

**Input:** $\mathcal{D}^{[1]} = \mathcal{D} = \{\boldsymbol{X}_i\}_{i=1}^N$, $N_L^{\text{tot}} = N_L$

**for** $l = 1, ..., N_L$ **do**

    Sample random basis functions $\boldsymbol{H}^{[l]}$ that satisfies Def. 5.2 starting from polynomials in level $l = 1$;

    Compute $\hat{f}^{[l]}(\boldsymbol{x})$ given $\mathcal{D}^{[l]}$ using Algorithm 1;

    $\mathcal{D}_{\text{mask}}^{[l]} \leftarrow \{\mathcal{D}^{[l]} \mid \hat{f}^{[l]}(\boldsymbol{X})/\hat{f}^{\text{hist}}(\boldsymbol{X}) > \boldsymbol{u}\}$ where $\boldsymbol{u} \sim \mathcal{U}([0,1])$;

    $m^{[l]} \leftarrow |\mathcal{D}_{\text{mask}}^{[l]}|/|\mathcal{D}|$;

    **if** $\sum_{j=1}^l |\mathcal{D}_{\text{mask}}^{[j]}| \approx |\mathcal{D}|$ **then**

        $\mathcal{D}_{\text{mask}}^{[l]} \leftarrow \mathcal{D}^{[l]}$;                                `// Mask all available samples`

        $m^{[l]} \leftarrow |\mathcal{D}_{\text{mask}}^{[l]}|/|\mathcal{D}|$;

        $N_L^{\text{tot}} \leftarrow l$;

        break;                                     `// Terminate the process`

    **else**

        $\mathcal{D}^{[l+1]} \leftarrow \mathcal{D}^{[l]} \backslash \mathcal{D}_{\text{mask}}^{[l]}$;               `// The uncovered samples are left for the next level`

    **end**

**end**

**Return** $\hat{f}(\boldsymbol{x}) = \sum_{l=1}^{N_L^{\text{tot}}} \hat{f}^{[l]}(\boldsymbol{x}) \, |\mathcal{D}_{\text{mask}}^{[l]}|/|D|$;       `// $N_L^{\text{tot}}$ is the total number of recursive calls`

---

# 7 Algorithm for MESSY estimation

The complete MESSY estimation algorithm is summarized in Algorithm 3. Within the iteration loop, following each application of the multi-level, symbolic, and recursive density recovery summarized in Algorithm 2, we introduce a maximum-cross entropy distribution (MxED) correction step (see Appendix B for details) to reduce any bias in our prediction for $\hat{f}$ from the former.

Finally, after completing the desired number of iterations, the algorithm returns the candidate density with the smallest KL Divergence given by

$$\mathrm{KL}\big(f \,\|\, \hat{f}\big) = \int f(\boldsymbol{x}) \log \left( \frac{f(\boldsymbol{x})}{\hat{f}(\boldsymbol{x})} \right) d\boldsymbol{x} \tag{26}$$

$$= - \int f(\boldsymbol{x}) \log \big( \hat{f}(\boldsymbol{x}) \big) d\boldsymbol{x} + \int f(\boldsymbol{x}) \log \big( f(\boldsymbol{x}) \big) d\boldsymbol{x} \tag{27}$$

$$\approx - \big\langle \log \big( \hat{f}(\boldsymbol{X}) \big) \big\rangle + \underbrace{\int f(\boldsymbol{x}) \log \big( f(\boldsymbol{x}) \big) d\boldsymbol{x}}_{\text{constant with respect to } \hat{f}} . \tag{28}$$

In other words, we use $- \big\langle \log \big( \hat{f}(\boldsymbol{X}) \big) \big\rangle$ as our selection criterion.

---

**Algorithm 3:** Pseudocode of the proposed `MESSY` estimation method. Here, $\boldsymbol{R}$ is the vector of linearly independent (polynomial) basis functions used in the moment matching procedure of MxED. Here, for MESSY-S the number of basis functions $N_b$ is sampled uniformly from the sample space $\Omega_{N_b}$, e.g. here we use $\Omega_{N_b} = \{2, ..., 8\}$ unless mentioned otherwise.

---

**Input:** $\mathcal{D} = \{\boldsymbol{X}_i\}_{i=1}^N$, $\Omega_{N_b}$, $N_m$, $N_{\text{iters}}$

Initialize $\hat{f}^{(i)} = 0$ for $i = 1, ..., N_{\text{iters}}$;

**for** $i = 1$ **to** $N_{\text{iters}}$ **do**

    **if** $i > 1$ **then**

        | Sample $N_b \sim \mathcal{U}(\Omega_{N_b})$;

    **end**

    Find $\hat{f}$ using multi-level, symbolic and recursive Algorithm 2 for density recovery;

    Generate samples of $\boldsymbol{Y} \sim \hat{f}$;

    Apply boundary condition (bounded/unbounded) to $\hat{f}$;

    Correct $\hat{f}$ using MxED (Algorithm 5) given samples $\boldsymbol{Y}$ as prior and $\mathbb{E}[\boldsymbol{R}(\boldsymbol{X})]$ as target moments;

    $\hat{f}^{(i)} \leftarrow \hat{f}$;

**end**

$\hat{f}^{\text{MESSY}-\text{P}} = \hat{f}^{(1)}$;

$\hat{f}^{\text{MESSY}-\text{S}} = \mathrm{argmin}_{\hat{f} \in \{\hat{f}^{(i)}\}_{i=1}^{N_{\text{iters}}}} \big( \mathrm{KL}(f \,\|\, \hat{f}) \big)$ ;

**Return** $\hat{f}^{\text{MESSY}-\text{P}}$ and $\hat{f}^{\text{MESSY}-\text{S}}$.

---

The MESSY algorithm comes in two flavors: MESSY-P, which considers only polynomial basis functions for $\boldsymbol{H}$, and MESSY-S which includes optimization over basis functions using the SR algorithms outlined above. In fact, by convention, the SR algorithm in MESSY-S starts its first iteration using polynomial basis functions up to order $N_m$ as the sample space of smooth functions. In other words, MESSY-P is a special case of MESSY-S with $N_{\text{iter}} = 1$. In the remaining iterations of MESSY-S, we perform the symbolic search in the space of smooth functions of order $N_m$ to find $N_b$ bases that provide manageable $\mathrm{cond}(\boldsymbol{L}^{\text{ME}})$, as discussed in Section 6.

In addition, we provide the option to enforce boundedness of $\hat{f}$ on the support that is specified by the user, i.e. letting $\hat{f}(\boldsymbol{x}) = 0$ for all $\boldsymbol{x}$ outside the domain of interest. This allows us to recover distributions with discontinuity at the boundary which may have application in image processing.

We also provide an option to further reduce the bias by minimizing the cross-entropy given samples of bounded/unbounded multi-level estimate as prior and moments of input samples as the target moments (see Appendix B for more details on the cross-entropy calculation). For this optional step, we generate samples of $\hat{f}$ and match the moments of polynomial basis functions up to order $N_m$. Since the solution at each level of $\hat{f}$ is close to the exact MED solution, the optimization problem associated with the moment matching procedure of the cross-entropy algorithm converges very quickly, i.e. in a few iterations, providing us with

a correction that minimizes bias along with the weighted samples of our estimate as the by-product. We note that in general the order of the randomly created basis function during the MESSY-S procedure may be different from the one used in the cross-entropy moment matching procedure.

## 8 Results

In this section we demonstrate the effectiveness of the proposed MESSY estimation method in recovering distributions given samples, using a number of numerical experiments, involving a range of distributions ranging from multi-mode to discontinuous. For validation, we provide comparisons with the standard KDE with an optimal bandwidth obtained using K-fold cross-validation with $K = 5$. We consider a uniform grid with 20 elements for the cross-validation in a range that spans from 0.01 to 10 on a logarithmic scale. We also compare with cross-entropy closure with Gaussian as the prior (MxED) using Newton's method. We note that while the standard maximum entropy distribution function differs from MxED as the latter incorporates a prior, we intentionally use MxED as a benchmark instead because the standard approach can be extremely expensive.

Unless mentioned otherwise, we report error, time, and KL Divergence by ensemble averaging over 25 for different sets of samples. Furthermore, in the case of MESSY-S we perform $N_{\text{iters}} = 10$ iterations, and we consider $(+, -, \times)$ operators and (cos, sin) functions. Here we report the execution time using a single-core CPU for each method. Typical symbolic expressions of density functions recovered by MESSY for the test cases considered here can be found in Appendix E. Furthermore, in Appendix C, we perform an ablation study that shows the importance of each component of the MESSY algorithm.

### 8.1 Bi-modal distribution function

For our first test case, we consider a one-dimensional bi-modal distribution function constructed by mixing two Normal distribution functions $\mathcal{N}(x \mid \mu, \sigma)$, i.e.

$$f(x) = \alpha \mathcal{N}(x \mid \mu_1, \sigma_1) + (1 - \alpha) \mathcal{N}(x \mid \mu_2, \sigma_2), \tag{29}$$

with $\alpha = 0.5$, means $\mu_1 = -0.6$ and $\mu_2 = 0.7$, and standard deviations $\sigma_1 = 0.3$ and $\sigma_2 = 0.5$.

Figure 2 compares results from MESSY, KDE and MxED for three different sample sizes, namely 100, 1000, and 10,000 samples of $f$. For MxED and MESSY-P, we use $N_b = N_m = 4$. In the case of MESSY-S, we randomly create $N_b$ basis functions which are $\mathcal{O}(x^4)$ (where $N_b$ is sampled uniformly within $\{2, \ldots, 8\}$). Both MESSY results are subject to a cross-entropy correction step with $N_b = 4$ polynomial moments. Clearly, the MxED and MESSY methods provide a reasonable estimate when a small number of samples is available; where KDE may suffer from bias introduced by the smoothing kernel. As a reference for the reader, we also compared the outcome density with a histogram in Appendix D.

In order to analyze the error further, Fig 3 presents the relative error in low and high order moments, KL Divergence, and single-core CPU time as the measure of computational cost for considered methods. Given optimal bandwidth is found, the KDE error can only be reduced by increasing the number of samples. However, maximum entropy based estimators such as MxED and MESSY provide more robust estimate when less samples are available. We point out that the convergence of the cases where only moments of polynomial basis functions are considered, i.e. MxED and MESSY-P, relies on the degree of the polynomials and not the number of samples. On the other hand, the additional search associated with MESSY-S returns more appropriate basis functions for a given upper bound on the order of the basis functions.

Next, we perform a convergence study on 10,000 samples and show that the parametric description converges to the solution when its degrees of freedom are increased. In Fig. 4, we show that both MESSY-P and MESSY-S converge to the true solution by increasing either the order of polynomial basis function, or the number of basis functions, respectively. In the case of MESSY-S, we generated symbolic expressions that are $\mathcal{O}(x^2)$. The improved agreement compared to the MESSY-P case highlights the benefit derived from non-traditional basis functions that may better represent the data.

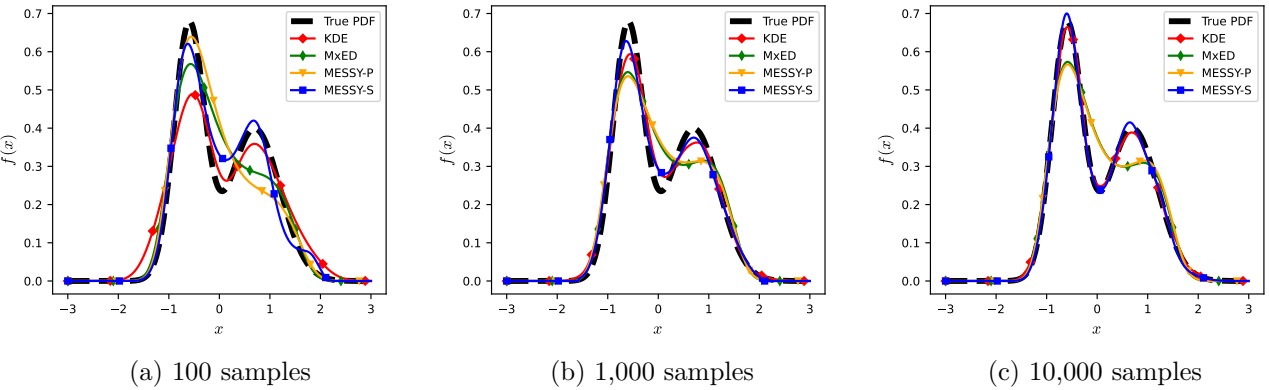

(a) 100 samples      (b) 1,000 samples      (c) 10,000 samples

Figure 2: Density estimation using KDE, MxED, MESSY-P, and MESSY-S given (a) 100, (b) 1,000, and (c) 10,000 samples.

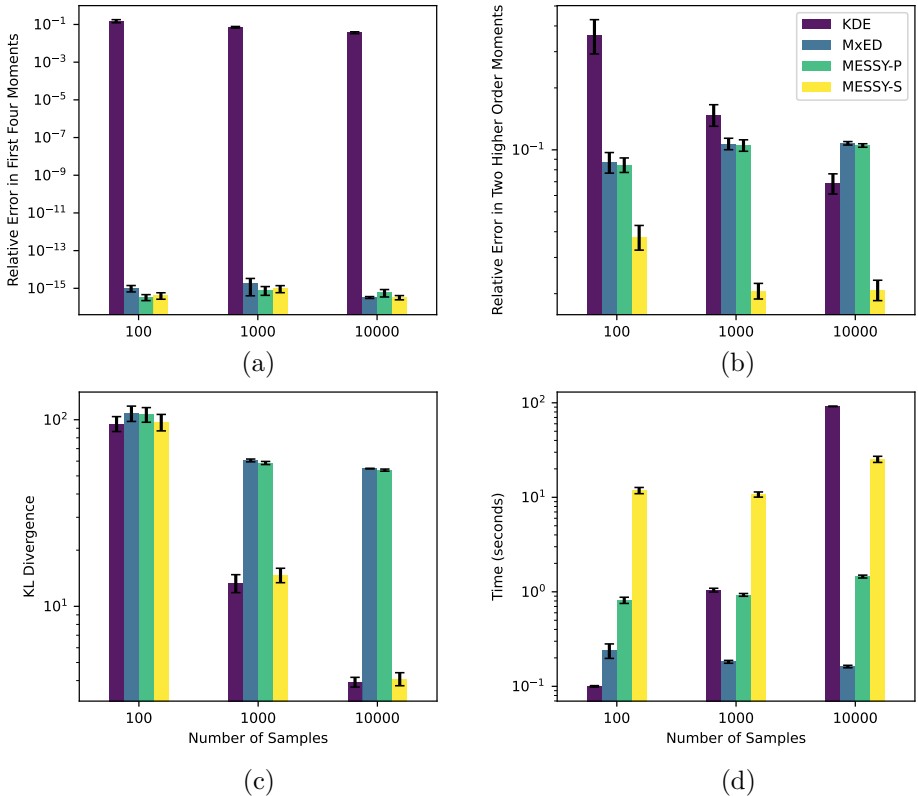

Figure 3: Comparing the relative error in (a) the first four moments, (b) two higher order moments (i.e. fifth and sixth moments), (c) KL Divergence, and (d) the execution time for KDE, MxED, MESSY-P, and MESSY-S in recovering distribution function for different sample sizes. Here, the error bar (in black) corresponds to the standard error of the empirical measurements.

As shown in Fig. 5, the MESSY-S procedure results in better-conditioned $L^{\mathrm{ME}}$ matrices than the MESSY-P for the same degrees of freedom. However, the search for a *good* basis function increases the computational cost. In each iteration of the search for basis functions, the MESSY-S algorithm may reject symbolic basis candidates based on the condition number of the matrix $L^{\mathrm{ME}}$. In other words, the improved performance associated with MESSY-S comes at some increased computational cost.

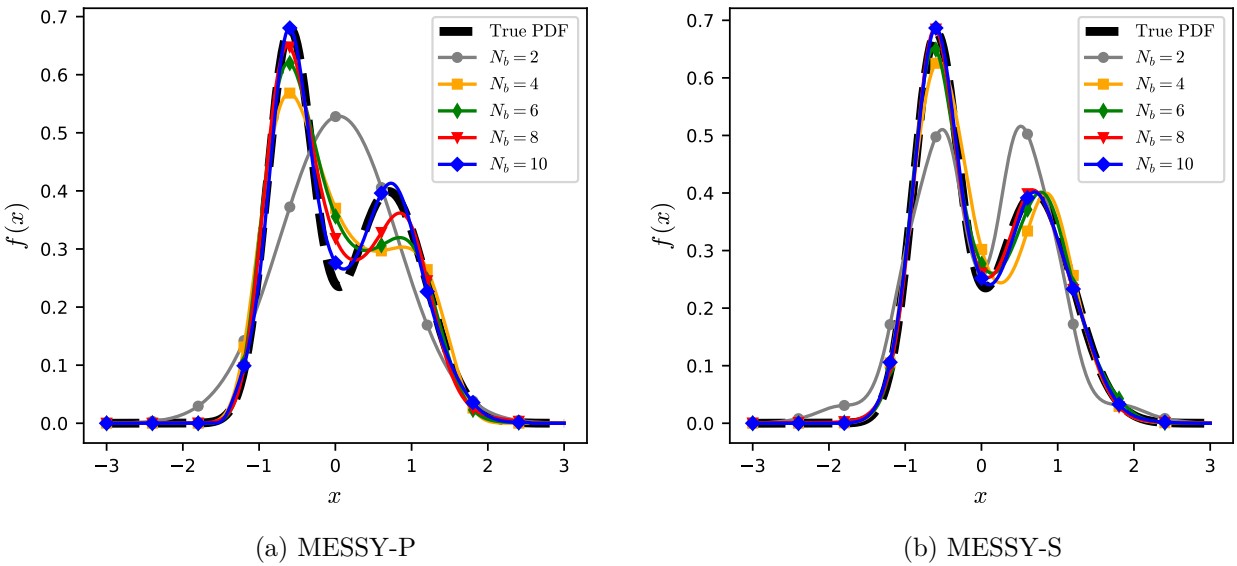

(a) MESSY-P         (b) MESSY-S

Figure 4: Convergence of MESSY estimation to target distribution function by (a) increasing the order of polynomial basis functions for MESSY-P or (b) increasing the number of randomly selected symbolic basis functions with $N_m = 2$ for MESSY-S.

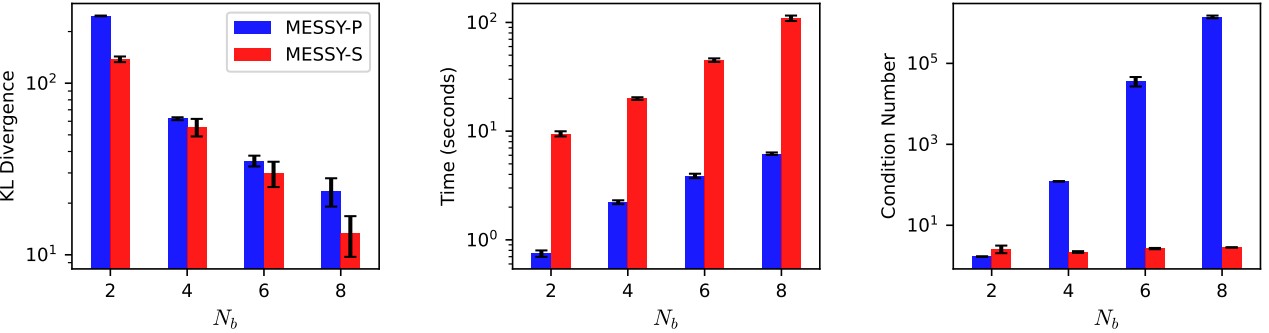

Figure 5: KL Divergence, execution time, and condition number against the degrees of freedom, i.e. the order of polynomial basis functions for MESSY-P or the number of symbolic basis functions with $N_m = 2$ for the MESSY-S estimate.

## 8.2 Limit of realizability

One of the challenging moment problems for maximum entropy methods is the one involving distributions near the border of physical realizability. In the one-dimensional case with moments of the first four monomials $[x, x^2, x^3, x^4]$ as the input, the moment problem is physically realizable when

$$\int x^4 f(x)dx \geq \left( \int x^3 f(x)dx \right)^2 + 1. \tag{30}$$

The moment problem with moments approaching the equality in Eq. (30) is called *limit of realizablity* (McDonald & Torrilhon, 2013; Akhiezer & Kemmer, 1965). We consider samples from a distribution in this limit as our test case here, since the standard MED cannot be solved due to an ill-conditioned Hessian (see Abramov (2007); Alldredge et al. (2014)).

In Fig. 6, we depict the estimated density of a bi-modal distribution in this limit given its samples with moments $\langle X \rangle = 0$, $\langle X^2 \rangle = 1$, $\langle X^3 \rangle = -2.10$ and $\langle X^4 \rangle = 5.42$. Here, we compare the density obtained

using KDE, MxED, MESSY-P, and MESSY-S to the histogram of samples. In this example, we obtained the MESSY-S estimate by searching in the space of smooth functions with $N_b \in \{2, ..., 8\}$ basis functions and compare polynomial and symbolic basis functions of order 2 and 4.

In Fig. 7, we compare the KL Divergence, the execution time, and the condition number for each method. While KDE suffers from over-smoothing and MxED/MESSY-P require at least $N_b = 4$ (and consequently $N_m = 4$, resulting in a stiff problem with large condition number), MESSY-S can obtain accurate density estimates by using unconventional basis functions with $N_m = 2$, thus maintaining a manageable condition number.

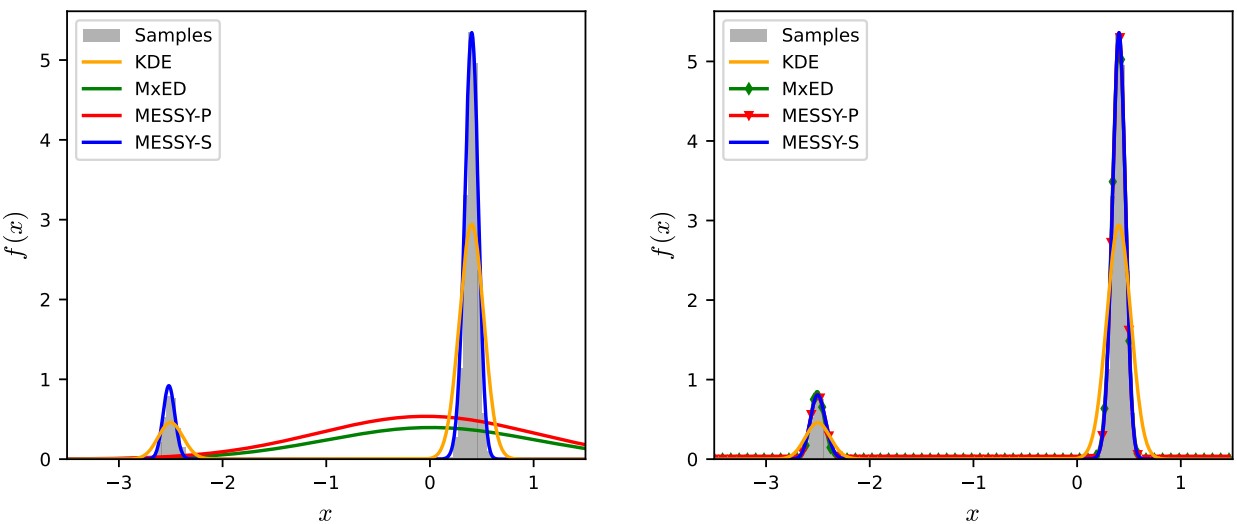

Figure 6: Estimating density for a case of distribution near the limit of realizability using KDE, MxED, MESSY-P, and MESSY-S. The solutions of MxED, MESSY-P, and MESSY-S are obtained using basis functions of second (left) and fourth (right) order.

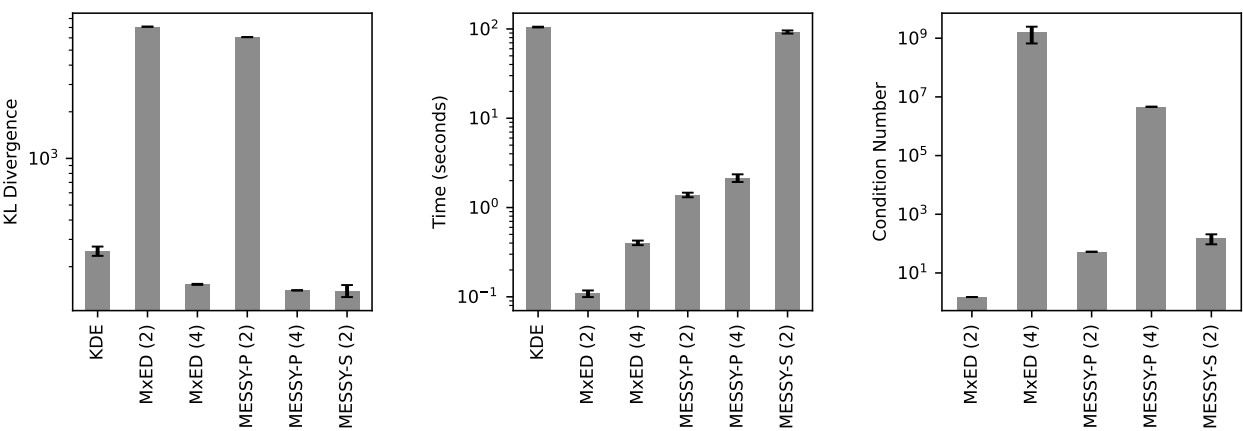

Figure 7: Comparing KL Divergence, execution time, and condition number of KDE, MxED, MESSY-P, and MESSY-S for an unknown distribution near the limit of the realizability. Here, we consider polynomial basis functions of second and fourth order for MxED and MESSY-P denoted by MxED (2), MxED (4), MESSY-P (2) and MESSY-P (4), respectively. In MESSY-S, we consider symbolic basis functions of second order only which we denote by MESSY-S (2).

### 8.3 Discontinuous distributions

We now highlight the benefits of using MESSY estimation with *piecewise continuous* capability for recovering distributions with a discontinuity at the boundary. As an example, let us consider the exponential distribution with a probability density function given by

$$f(x) = \begin{cases} ae^{-ax} & \text{if } x \geq 0 \\ 0 & \text{otherwise} \end{cases} \tag{31}$$

with $a = 1$.

Given $10,000$ samples of this distribution, in Fig. 8 we compare KDE, MxED, and the proposed MESSY-P and MESSY-S methodologies. In the case of MxED and MESSY-P we consider second-order polynomial basis functions, and for MESSY-S we search the space of smooth functions for $N_b \in \{2, ..., 8\}$ symbolic basis functions of order $\mathcal{O}(x^2)$. For MESSY-P and MESSY-S, we apply the boundary condition

$$\hat{f}(x) = 0 \quad \forall x < \min(X). \tag{32}$$

By providing information about the boundedness of the expected distribution, we enable MESSY to accurately predict densities with discontinuity near the boundary. As it can be seen clearly from Fig. 8, in contrast to MxED, both MESSY-P and MESSY-S provide accurate predictions by taking advantage of the information about the boundedness of the target density.

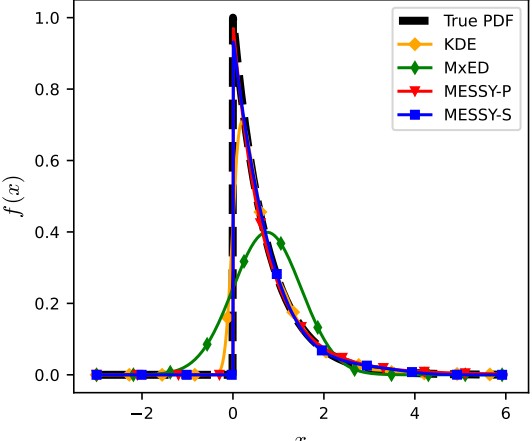

Figure 8: Estimating density of exponential distribution function from its samples using KDE, MxED, MESSY-P, and MESSY-S. For MxED, MESSY-P and MESSY-S, with $N_m = 2$.

The KL Divergence score and execution time for each method is shown in Fig. 9. These figures show that MESSY-P and MESSY-S provide a more accurate description compared to the KDE estimate, albeit at a higher computational cost.

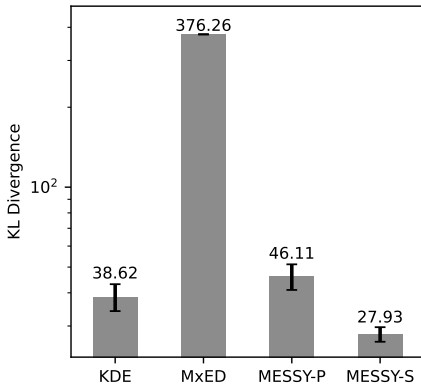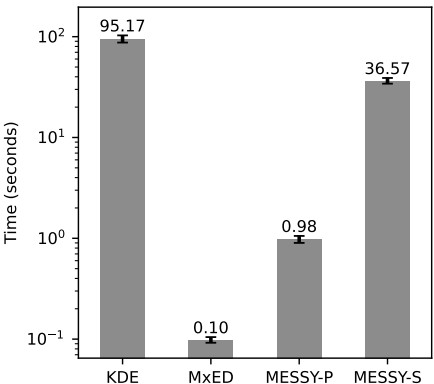

Figure 9: KL Divergence and execution time for KDE, MxED, MESSY-P, and MESSY-S estimation of exponential distribution function given 10,000 samples.

### 8.4 Two-dimensional distributions

In this section the proposed MESSY methodology is applied to the estimation of 2-dimensional distributions. Let us consider a multivariate Gaussian distribution in two dimensions, i.e. $\boldsymbol{X} = (X_1, X_2) \sim \mathcal{N}(\boldsymbol{\mu}, \boldsymbol{\Sigma})$ with probability density

$$\mathcal{N}(\boldsymbol{x}; \boldsymbol{\mu}, \boldsymbol{\Sigma}) = \frac{1}{(2\pi)^{d/2}\sqrt{\det(\boldsymbol{\Sigma})}} \exp\left\{-\frac{1}{2}(\boldsymbol{x} - \boldsymbol{\mu})^T \boldsymbol{\Sigma}^{-1}(\boldsymbol{x} - \boldsymbol{\mu})\right\} \tag{33}$$

where $d = 2$, $\boldsymbol{x} = (x_1, x_2) \in \mathbb{R}^2$, $\boldsymbol{\mu}$ is the mean vector, and $\boldsymbol{\Sigma}$ is the covariance matrix. In this example,

$$\boldsymbol{\mu} = \begin{bmatrix} 0 \\ 0 \end{bmatrix}, \qquad \boldsymbol{\Sigma} = \begin{bmatrix} 1 & 0.5 \\ 0.5 & 1 \end{bmatrix}. \tag{34}$$

Let us also consider a more complex 2D distribution by multiplying two independent 1D distributions, namely the exponential and Gamma distributions, i.e. $X_1 \sim \text{Exp}(\lambda)$, $X_2 \sim \Gamma(k, \theta)$, and $\boldsymbol{X} = (X_1, X_2) \sim \text{Exp}(\lambda)\Gamma(k, \theta)$. In particular, we consider samples of the joint pdf given by

$$f(x_1, x_2; \lambda, k, \theta) = f(x_1; \lambda)f(x_2; k, \theta) = \begin{cases} \lambda e^{-\lambda x_1} \cdot \frac{x_2^{k-1} e^{-x_2/\theta}}{\Gamma(k)\theta^k} & x_1 \geq 0, \ x_2 \geq 0 \\ 0 & \text{otherwise} \end{cases} \tag{35}$$

where $\lambda > 0$ is the rate parameter, $k > 0$ is the shape parameter, $\theta > 0$ is the scale parameter, and $\Gamma(\cdot)$ is the gamma function. In our example, we use $\lambda = 2$, $k = 3$, and $\theta = 0.5$.

Fig. 10 compares the true distribution against the KDE, MESSY-P and MESSY-S estimate given $10^4$ samples of the distribution. In cases of MESSY-P and MESSY-S, we consider basis functions up to 2nd order in each dimension. The figure shows that KDE, MESSY-P and MESSY-S provide a reasonable estimate for the multivariate normal distribution. However, among all considered estimators for the case of the Gamma-exponential distribution, MESSY-S seems to provide a slightly better estimate, i.e. it captures the peak in the most probable region of the distribution.

### 8.5 Cost of MESSY-P with respect to dimension

As discussed in section 4.1, for a given moment problem (fixed vector of basis functions), the cost of evaluating Lagrange multipliers in the proposed method is linear with respect to number of samples and independent of dimension — a benefit of Monte Carlo integration. However, similar to the standard MED approach, the proposed MESSY estimate requires the inversion of a Hessian matrix $\boldsymbol{L} \in \mathbb{R}^{N_b \times N_b}$. Therefore, in addition to integration, a MESSY cost estimate also includes solution of a linear system of equations with

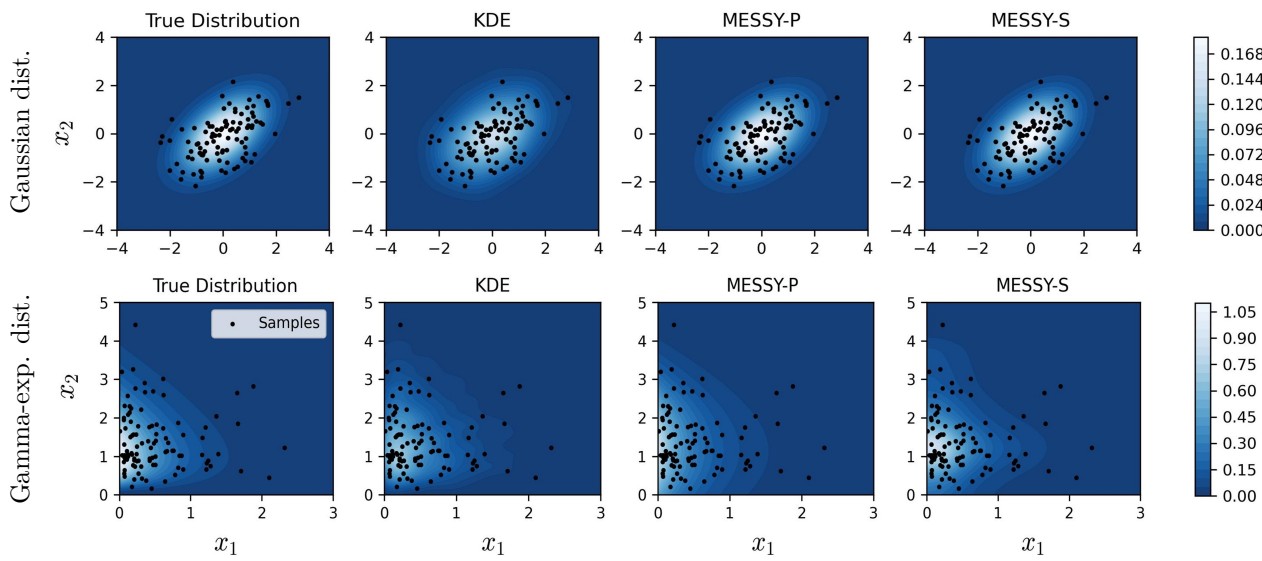

Figure 10: Estimating the density of samples in two dimensions using KDE, MESSY-P, and MESSY-S. We use 10,000 samples from a Gaussian (top) and the Gamma-exponential density defined in Eq. (35) (bottom). For better visualization, we only show 100 random samples. We also report that the KL-divergence for MESSY-S, MESSY-P and KDE are on the same order.

cost of order $\mathcal{O}(N_b^3)$. In this example, we consider the MESSY-P estimator with basis functions $\boldsymbol{H} = \left[x_1, ..., x_d, x_1^2, x_1 x_2, ..., x_d^2\right]$ for estimating a target $d$-dimensional multivariate normal distribution Eq. (33) with mean $\boldsymbol{\mu}$

$$\mu \sim \mathcal{N}(\boldsymbol{0}, \frac{1}{2}\boldsymbol{I}) \tag{36}$$

and covariance matrix $\boldsymbol{\Sigma}$

$$\boldsymbol{\Sigma} = \boldsymbol{I} + \boldsymbol{U}\boldsymbol{U}^T \tag{37}$$

$$\text{where} \quad U_{i,j} = \begin{cases} 0 & i < j \\ \xi & \text{otherwise} \end{cases} \quad \text{and} \quad \xi \sim \mathcal{N}(0, \frac{1}{2}) . \tag{38}$$

Using 2nd order polynomial basis functions, the number of unique basis functions is $N_b = d + d(d+1)/2$, leading to a cost of order $\mathcal{O}((d + d(d+1)/2)^3)$ for a $d$-dimensional probability space. In Fig. 11-(a), we illustrate this by showing the normalized execution time (normalized by the time for $d = 1$ and given sample size) against the dimension of the target density given $N \in \{100, 200, 400, 800, 1600, 3200\}$ samples. Furthermore, in order to highlight the linear dependence of cost with number of samples $N$ for a fixed moment problem, in Fig. 11-(b) we also report the normalized execution time of computing Lagrange multipliers (normalized by the time for $N = 100$ and dimension $d$ of target density) as a function of number of samples.

Since the most expensive component of MESSY, i.e. the computation of Lagrange multipliers, has been carried out on a single computer core with a direct solver, we confirm that MESSY algorithm can be used for density recovery in large number of dimensions at a feasible computational cost, regardless of the scaling with dimension. The limiting factor appears to be the storage associated with solving the linear system. In case a larger system than the one studied here are of interest, one may use iterative solvers, see Saad (2003).

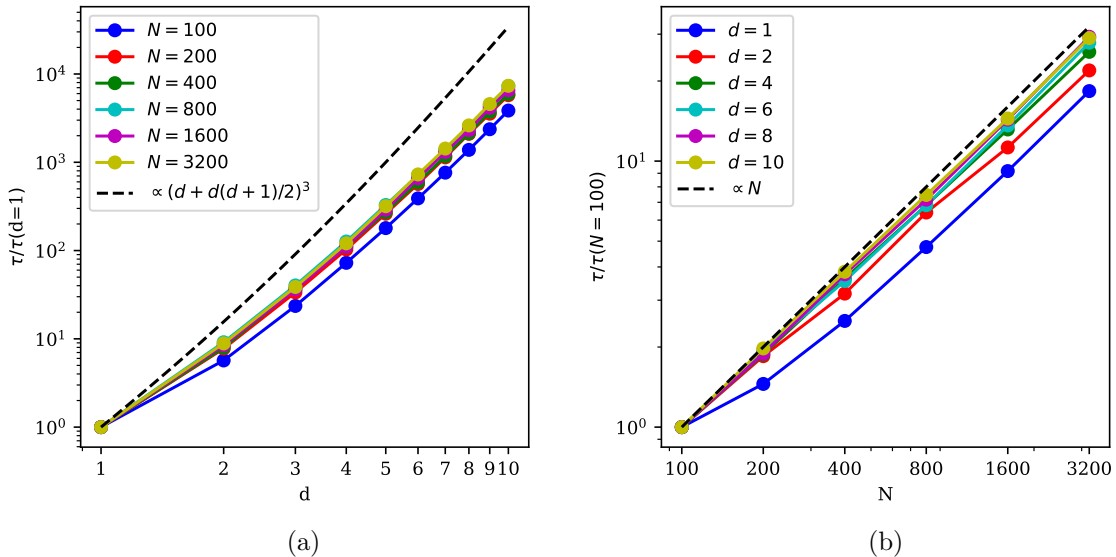

Figure 11: Relative execution time $\tau$ of computing Lagrange multipliers given $N \in \{100, 200, 400, 800, 1600, 3200\}$ samples of $d$-dimensional multivariate normal distribution function where $d = 1, ..., 10$. Left (a) shows the normalized execution time versus dimension and right (b) the normalized execution time versus number of samples. The execution time is computed on a single core and single thread of 2.3GHz Quad-Core Intel Core i7 processor and averaged over 5 ensembles.

## 9 Conclusion and Outlook

We present a new method for symbolically recovering the underlying probability density function of a given finite number of its samples. The proposed method uses the maximum entropy distribution as an ansatz thus guaranteeing positivity and least bias. We devise a method for finding parameters of this ansatz by considering a Gradient flow in which the ansatz serves as the driving force. One main takeaway from this work is that the parameters of the MED ansatz can be computed efficiently by solving a linear problem involving moments calculated from the given samples. We also show that the complexity associated with finding Lagrange multipliers, the most expensive part of the algorithm, scales linearly with the number of samples in all dimensions. On the other hand, the cost of inverting the Hessian matrix with dimension $\mathbb{R}^{N_b \times N_b}$ is $\mathcal{O}(N_b^3)$, where the number of basis functions $N_b$ grows with dimension. Overall, our numerical experiments show that densities of dimension 10 can be recovered using a single core of a typical workstation. It is worth noting that since the whole probability space is represented with a few parameters, the MESSY representation can reduce the training cost associated with the PDE-FIND Rudy et al. (2017) and SINDy Brunton et al. (2016) method, where in the latter the regression is done on a data set that discretizes the whole space with a large number of parameters that need to be fit. Further detailed analysis of the trade-off between cost and accuracy in inferring densities in high-dimensional probability spaces will be addressed in future work.

The second main takeaway from this work is that accurate density recovery does not necessarily require the use of high-order moments. In fact, increasing the number of complex but low-order basis functions leads to superior expressiveness and better assimilation of the data. For this reason, the proposed method is equipped with a Monte Carlo search in the space of smooth functions for finding basis functions to describe the exponent of the MED ansatz, using KL Divergence, calculated from the unknown-distribution samples, as an optimality criterion. Discontinuous densities are treated by considering piece-wise continuous functions with support on the space covered by samples.

We validate and test the proposed MESSY estimation approach against benchmark non-parametric (KDE) and parametric (MxED) density estimation methods. In our experiments, we consider three canonical test cases; a bi-modal distribution, a distribution close to the limit of realizability, and a discontinuous

distribution function. Our results suggest that MESSY estimation exhibits several positive attributes compared to existing methods. Specifically, while retaining some of the most desirable features associated with MED, namely non-negativity, least bias, and matching the moments of the unknown distribution, it outperforms standard maximum-entropy-based approaches for two reasons. First, it uses samples of the target distribution in the evaluation of the Hessian, which has a linear cost with respect to the dimension of the random variable. Second, the resulting linear problem for finding the Lagrange multipliers from moments is significantly more efficient than the Newton line search used by the classical MED approach. Moreover, our multi-level algorithm allows for recovery of more complex distributions compared to the standard MED approach. Combining the efficient approach of finding maximum entropy density via a linear system with the symbolic exploration for the optimal basis functions paves the way for achieving low bias, consistent, and expressive density recovery from samples.

Although the proposed MESSY estimate alleviates a number of numerical challenges associated with finding the MED given samples, it cannot expand the space of the MED solution. In other words, if the moment problem does not permit any MED solutions, e.g. (Junk, 1998), the MESSY estimate of MED fails to infer any densities since there is no MED solution to be found. Secondly, there are no guarantees that the coefficient of the leading term in the exponent of the MED will be negative. Hence, in unbounded domains the MESSY estimate may diverge in the tails of the distribution. Both of these issues can be resolved by regularizing the MED ansatz, e.g. via incorporating a Wasserstein distance from a prior distribution as suggested by Sadr et al. (2023).

The most important, perhaps, distinguishing characteristic of the proposed methodology from non-parametric approaches, such as KDE, is the ability to recover a tractable symbolic estimator of the unknown density. This can be beneficial in a number of applications, such as, for example, finding the underlying dynamics of a stochastic process. In such a case, where it is known that the transition probability takes an exponential form, this method can be used for recovering the drift and diffusion terms of an Ito process given a sequence of random variables, see e.g. Risken (1989); Oksendal (2013). Furthermore, the proposed method may be used to find a relation between moments of interest which could be helpful in determining the parameters of a statistical model, such as a closure for a hierarchical moment description of fluid flow far from equilibrium. That said, exploring the efficiency of our algorithm in high-dimensional contexts is a critical next step, considering the complexity of modern machine learning datasets. Other possible directions for future work include: (i) data-driven discovery of governing dynamical laws from samples; and (ii) applications to variance reduction.

### Acknowledgments

We would like to acknowledge Prof. Youssef Marzouk and Evan Massaro for their helpful comments and suggestions. M. Sadr thanks Hossein Gorji for his stimulating and constructive comments about the core ideas presented in this paper. T. Tohme was supported by Al Ahdab Fellowship. M. Sadr acknowledges funding provided by the German Research Foundation (DFG) under grant number SA 4199/1-1.

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

# A    Maximum entropy distribution function

The maximum entropy distribution (MED) function finds the least-biased closure for the moment problem. Given $N_b$ realizable moments $\boldsymbol{\mu} \in \mathbb{R}^{N_b}$ associated with polynomial basis functions $\boldsymbol{H}$ of the unknown distribution function, MED is obtained by minimizing the Shannon entropy with constraint moments using the method of Lagrange multipliers as Hauck et al. (2008)

$$
C[\mathcal{F}(\boldsymbol{x})] := \int \mathcal{F}(\boldsymbol{x})(\log\left(\mathcal{F}(\boldsymbol{x})\right) - 1)d\boldsymbol{x} - \sum_{i=1}^{N_b} \lambda_i \left( \int H_i(\boldsymbol{x})\mathcal{F}(\boldsymbol{x})d\boldsymbol{x} - \mu_i(\boldsymbol{x})\right) \ . \tag{A.1}
$$

By taking the variational derivative of functional A.1, the extremum is found as

$$
\mathcal{F}(\boldsymbol{x}) = \frac{1}{Z}\exp\left(\sum_{i=1}^{N_b} \lambda_i H_i(\boldsymbol{x})\right), \qquad \text{where} \ \ Z = \int \exp\left(\sum_{i=1}^{N_b} \lambda_i H_i(\boldsymbol{x})\right)d\boldsymbol{x}, \tag{A.2}
$$

which is referred to as the maximum entropy distribution function. The Lagrange multipliers $\boldsymbol{\lambda}$ appearing in Eq. (A.2) can be found using the Newton-Raphson approach. As formulated in (Debrabant et al., 2017), the unconstrained dual formulation $D(\boldsymbol{\lambda})$ provides us with the gradient $\boldsymbol{g} = \nabla D(\boldsymbol{\lambda})$ and Hessian $\boldsymbol{L}(\boldsymbol{\lambda}) = \nabla^2 D(\boldsymbol{\lambda})$ as

$$
g_i = \mu_i - \frac{1}{Z}\int H_i \exp\left(\sum_{k=1}^{N_b} \lambda_k H_k\right)d\boldsymbol{x} \quad \text{for } i = 1,...,N_b \tag{A.3}
$$

$$
\text{and} \quad L_{i,j} = -\frac{1}{Z}\int H_i H_j \exp\left(\sum_{k=1}^{N_b} \lambda_k H_k\right)d\boldsymbol{x} \quad \text{for } i,j = 1,...,N_b \ . \tag{A.4}
$$

Once the gradient and Hessian are computed, the Lagrange multipliers $\boldsymbol{\lambda}$ can be updated via

$$
\boldsymbol{\lambda} \leftarrow \boldsymbol{\lambda} - \boldsymbol{L}^{-1}(\boldsymbol{\lambda})\boldsymbol{g}(\boldsymbol{\lambda}) \tag{A.5}
$$

as detailed in Algorithm 4.

---

**Algorithm 4:** Newton's method for finding Lagrange multipliers of MED given moments $\boldsymbol{\mu}$ for a given tolerance $\epsilon$.

---

**Input:** $\boldsymbol{\mu}$, $\boldsymbol{\lambda}_0$
Initialize $\boldsymbol{\lambda} \leftarrow \boldsymbol{\lambda}_0$;
Compute gradient $\boldsymbol{g}$ and Hessian $\boldsymbol{L}$, i.e. Eqs. (A.3)-(A.4);
**while** $||\boldsymbol{g}|| > \epsilon$ **do**
    Update $\boldsymbol{\lambda} \leftarrow \boldsymbol{\lambda} - \boldsymbol{L}^{-1}\boldsymbol{g}$;
    Update gradient $\boldsymbol{g}$ and Hessian $\boldsymbol{L}$ with the new $\boldsymbol{\lambda}$ via numerical integration of Eqs. (A.3)-(A.4);
**end**
**Return** $\boldsymbol{\lambda}$;

---

## B    Maximum cross-entropy distribution function

The maximum cross-entropy distribution function (MxED) finds the least-biased closure for the moment problem given $N_b$ realizable moments $\boldsymbol{\mu} \in \mathbb{R}^{N_b}$ associated with polynomial basis functions $\boldsymbol{H}$ of the unknown distribution function along with the prior $\mathcal{F}^{\mathrm{Prior}}$ as the input. In other words, in addition to the moments of the target distribution, in the MxED method we also have access to a prior distribution $\mathcal{F}^{\mathrm{Prior}}$ as well as $N$ samples of the former, i.e. $\{\boldsymbol{X}_j^{\mathrm{prior}}\}_{j=1}^N \sim \mathcal{F}^{\mathrm{Prior}}$. MxED is obtained by minimizing the Shannon cross-entropy from the prior with constraint on moments using the method of Lagrange multipliers via the functional

$$C[\mathcal{F}(\boldsymbol{x})] := \int \mathcal{F}(\boldsymbol{x}) \log\left(\frac{\mathcal{F}(\boldsymbol{x})}{\mathcal{F}^{\mathrm{Prior}}(\boldsymbol{x})}\right) d\boldsymbol{x} + \sum_{i=1}^{N_b} \lambda_i \left(\int H_i(\boldsymbol{x})\mathcal{F}(\boldsymbol{x})d\boldsymbol{x} - \mu_i(\boldsymbol{x})\right) . \tag{B.1}$$

By taking the variational derivative of functional B.1, the extremum is found to be

$$\mathcal{F}(\boldsymbol{x}) = \frac{1}{Z}\mathcal{F}^{\mathrm{Prior}}(\boldsymbol{x})\exp\left(\sum_{i=1}^{N_b} \lambda_i H_i(\boldsymbol{x})\right), \qquad \text{where } Z = \int \mathcal{F}^{\mathrm{Prior}}(\boldsymbol{x})\exp\left(\sum_{i=1}^{N_b} \lambda_i H_i(\boldsymbol{x})\right) d\boldsymbol{x}. \tag{B.2}$$

Similar to the maximum entropy distribution function, the Lagrange multipliers $\boldsymbol{\lambda}$ appearing in Eq. (B.2) can be found by following the Newton-Raphson approach. As formulated in (Debrabant et al., 2017), the unconstrained dual formulation $D(\boldsymbol{\lambda})$ provides us with the gradient $\boldsymbol{g} = \nabla D(\boldsymbol{\lambda})$ and Hessian $\boldsymbol{L}(\boldsymbol{\lambda}) = \nabla^2 D(\boldsymbol{\lambda})$ as

$$g_i = \mu_i - \frac{1}{Z}\int \mathcal{F}^{\mathrm{Prior}} H_i \exp\left(\sum_{k=1}^{N_b} \lambda_k H_k\right) d\boldsymbol{x} \quad \text{for } i = 1, ..., N_b \tag{B.3}$$

$$\text{and} \quad L_{i,j} = -\frac{1}{Z}\int \mathcal{F}^{\mathrm{Prior}} H_i H_j \exp\left(\sum_{k=1}^{N_b} \lambda_k H_k\right) d\boldsymbol{x} \quad \text{for } i,j = 1, ..., N_b . \tag{B.4}$$

Once the gradient and Hessian are computed, the Lagrange multipliers $\boldsymbol{\lambda}$ can be updated via

$$\boldsymbol{\lambda} \leftarrow \boldsymbol{\lambda} - \boldsymbol{L}^{-1}(\boldsymbol{\lambda})\boldsymbol{g}(\boldsymbol{\lambda}) . \tag{B.5}$$

Since we have access to the samples of $\boldsymbol{X}^{\mathrm{Prior}} \sim \mathcal{F}^{\mathrm{Prior}}$, we use the given samples to compute the gradient and Hessian, i.e.

$$g_i \approx \mu_i - \left\langle H_i\big(\boldsymbol{X}^{\mathrm{Prior}}\big)\right\rangle_W \quad \text{for } i = 1, ..., N_b \tag{B.6}$$

$$L_{i,j} \approx -\left\langle H_i\big(\boldsymbol{X}^{\mathrm{Prior}}\big)H_j\big(\boldsymbol{X}^{\mathrm{Prior}}\big)\right\rangle_W \quad \text{for } i,j = 1, ..., N_b , \tag{B.7}$$

where $W(\boldsymbol{X}^{\mathrm{Prior}}) = \exp\left(\sum_{k=1}^{N_b} \lambda_k H_k\big(\boldsymbol{X}^{\mathrm{Prior}}\big)\right)$ denotes weights for calculating moments using importance sampling, i.e. $\langle\phi(\boldsymbol{X})\rangle_W := \sum_{j=1}^N \phi(\boldsymbol{X}_j)W(\boldsymbol{X}_j)/\sum_{j=1}^N W(\boldsymbol{X}_j)$. More details can be found below (Algorithm 5).

---

**Algorithm 5:** Newton's method for finding Lagrange multipliers of MxED given moments $\boldsymbol{\mu}$ and samples of prior $\boldsymbol{X}^{\mathrm{Prior}} \sim \mathcal{F}^{\mathrm{Prior}}$ for a given tolerance $\epsilon$.

---

**Input:** $\boldsymbol{\mu}$, $\boldsymbol{X}^{\mathrm{Prior}} \sim \mathcal{F}^{\mathrm{Prior}}$
Initialize $\boldsymbol{\lambda} \leftarrow \boldsymbol{0}$;
Compute gradient $\boldsymbol{g}$ and Hessian $\boldsymbol{L}$, i.e. Eqs. (B.6)-(B.7);
**while** $\|\boldsymbol{g}\| > \epsilon$ **do**
 | Update $\boldsymbol{\lambda} \leftarrow \boldsymbol{\lambda} - \boldsymbol{L}^{-1}\boldsymbol{g}$;
 | Update gradient $\boldsymbol{g}$ and Hessian $\boldsymbol{L}$ with the new $\boldsymbol{\lambda}$ using samples, i.e. Eq. (B.6)-(B.7);
**end**
**Return** $\boldsymbol{\lambda}$;

---

## C   Ablation studies

In this section, we report on the results of an ablation study that shows the effects of each component in the proposed MESSY solution. The four components of MESSY Estimation are: i) the symbolic component, ii) the multi-level component, iii) the orthogonalization, and iv) the cross-entropy step. As we have already compared MESSY-S and MESSY-P (MESSY without the symbolic regression component) throughout the paper, we conduct an ablation study on the remaining components, i.e. the multi-level, the orthogonalization, and the cross-entropy step.

### C.1   Multi-level component of MESSY

First, in order to show the effect of the multi-level part of the MESSY algorithm, let us consider the bi-modal density near the limit of realizability, see 8.2. In particular, let us consider the MESSY-P estimate with 4th order polynomials as basis functions. In Fig. 12, we show that by disabling the multi-level step, the estimated density covers only one of the peaks and entirely misses the other one. We believe that this is due to the high conditionality of the matrix $\boldsymbol{L}^{\mathrm{ME}}$. However, MESSY-P with a multi-level step accurately recovers both peaks of the underlying density.

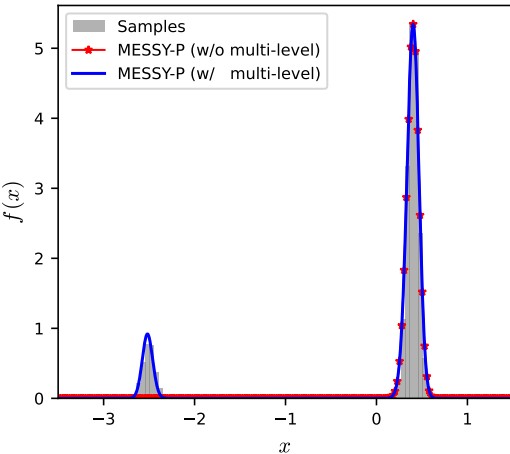

Figure 12: Ablation study on the multi-level component of MESSY-P estimation for the case of target distribution near the limit of realizability.

### C.2   Orthogonalization

Another important element of MESSY process is the orthogonalization of basis functions given samples, i.e. the modified Gram-Schmidt Algorithm 1. As an example, here we consider the MESSY-P estimate of target bimodal distribution function presented in 8.1 as a test case and report the condition number of the matrix $\boldsymbol{L}^{\mathrm{ME}}$ that needs to be inverted to compute the Lagrange multipliers in Eq. (18). As shown in Fig. 13, the condition number of $\boldsymbol{L}^{\mathrm{ME}}$ increases with the polynomial order, making the linear system (18) stiff and difficult to solve. However, by orthogonalizing the basis functions, we can maintain a low condition number for the outcome $\boldsymbol{L}^{\mathrm{ME}}$ which makes the resulting linear system tractable.

### C.3   Cross-entropy step

The final step in the MESSY-P procedure is the cross-entropy. The goal of this step is to reduce the bias that may have been introduced by the multi-level component of the algorithm. In the MESSY Algorithm 3, we take the outcome of the multi-level step as the prior, and correct Lagrange multipliers to match the target moments. In Fig. 14, we compare the MESSY-P density estimate of the discontinuous density from the test case presented in 8.3 with and without cross-entropy step. As expected, we observe that the cross-entropy

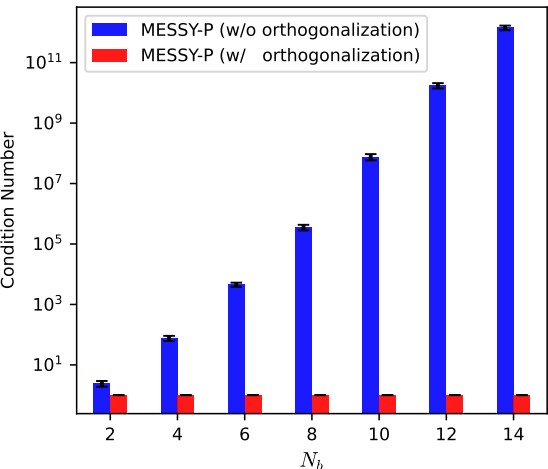

Figure 13: Ablation study on the orthogonalization component of MESSY-P algorithm for the case of bimodal distribution. Relation between condition number of matrix $\boldsymbol{L}^{\mathrm{ME}}$ and the order of considered polynomial with and without the orthogonalization step.

step is essential in recovering distributions with discontinuities. This is due to the fact that the solution obtained via gradient flow assumes a smooth target density. Hence, it fails to find the Lagrange multipliers for the target discontinuous pdf. However, the cross entropy step only assumes that the target distribution is integrable to the extent that the given moments exist. This weaker condition is essential to recovering discontinuous distributions, e.g. the one showed in Fig. 14.

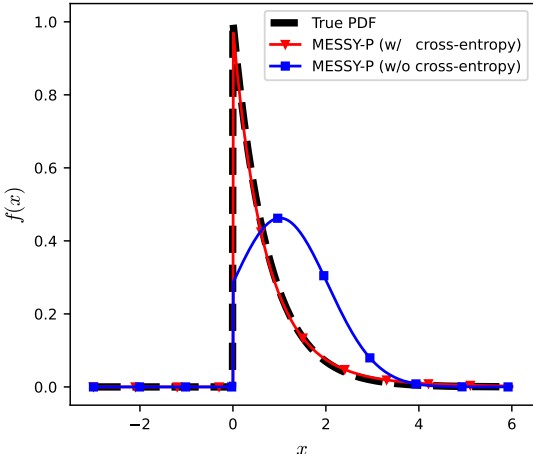

Figure 14: Ablation study on the cross-entropy step of MESSY-P estimation of a discontinuous density.

## D  Histogram estimate to the bi-modal distribution function

Figure 15 provides a comparison between a histogram, KDE, MESSY-P and MESSY-S estimates of the underlying bi-modal distribution function considered in 8.1 given $N = 100, \ 1000, 10000$ samples. Clearly, MED estimates provide a better solution than non-parametric ones, i.e. histogram and KDE, when not many samples are available. However, as more samples are considered, the non-parametric estimators become more accurate than MESSY (or any parametric estimator) with fixed number of basis functions.

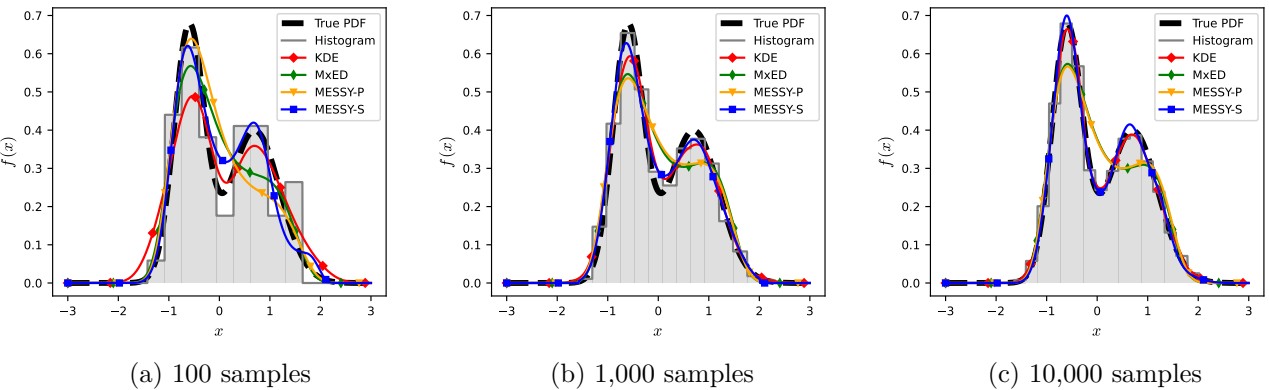

(a) 100 samples       (b) 1,000 samples       (c) 10,000 samples

Figure 15: Density estimation using KDE, MxED, MESSY-P, MESSY-S and histogram given (a) 100, (b) 1,000, and (c) 10,000 samples.

# E   Solution found by MESSY for the considered test cases

Table 1: Density expressions recovered by our MESSY estimation method for several distributions.

| Example | | Expression |
|---|---|---|
| Bimodal 1D | MESSY-P | $\hat{f}(x) = 0.288e^{-0.017x^{10}+0.106x^9-0.084x^8-0.659x^7+1.209x^6+1.179x^5-3.722x^4+0.075x^3+2.693x^2-0.612x}$ |
| | MESSY-S | $\hat{f}(x) = 0.993e^{-1.85x^2-1.162x\cos(1.5x)+0.232x-0.652\cos(x)-0.424\cos(2x)-0.591\cos(3.5x)+0.47\cos(\cos(3.5x))}$ |
| Limit of | MESSY-P | $\hat{f}(x) = 1.591 \cdot 10^{-6}e^{-12.876x^4-56.46x^3-38.072x^2+62.617x} + 5.282 \cdot 10^{-27}e^{-7.969x^4-28.862x^3-4.342x^2+20.938x}$ |
| Realizability | MESSY-S | $\hat{f}(x) = 4.134 \cdot 10^{81}e^{-21.893x^2\sin(x)+0.025x^2+117.267x\cos(x)+0.861x+395.584\sin^2(x)-57.393\sin(x)+200.421\cos(x)-744.874\cos(\cos(x))}$ |
| Discontinuous | MESSY-P | $\hat{f}(x) = \begin{cases} 1.096\,e^{0.086x^2-1.298x} & \text{if } x \geq 0 \\ 0 & \text{otherwise} \end{cases}$ |
| | MESSY-S | $\hat{f}(x) = \begin{cases} 0.293\,e^{-0.145x^2+0.018x+0.251\cos(x)\cos(1.5x)+0.713\cos(x)+0.09\cos(1.5x)\cos(3x)+0.076\cos(3.5x)} & \text{if } x \geq 0 \\ 0 & \text{otherwise} \end{cases}$ |
| Gaussian 2D | MESSY-P | $\hat{f}(x_1, x_2) = 0.18e^{-0.606x_1^2+0.572x_1x_2+0.029x_1-0.663x_2^2-0.075x_2}$ |
| | MESSY-S | $\hat{f}(x_1, x_2) = 0.182e^{-0.648x_1^2+0.598x_1x_2-0.018x_1-0.644x_2^2-0.044x_2}$ |
| Gamma-exp | MESSY-P | $\hat{f}(x_1, x_2) = \begin{cases} 0.477e^{0.225x_1^2-0.04x_1x_2-2.264x_1-0.376x_2^2+0.913x_2} & x_1 \geq 0,\, x_2 \geq 0 \\ 0 & \text{otherwise} \end{cases}$ |
| | MESSY-S | $\hat{f}(x_1, x_2) = \begin{cases} 0.017e^{-0.13x_1^2+0.016x_1x_2+0.15x_1+0.017x_2^2\cos(2.5\cos(2x_2))-0.37x_2^2+0.858x_2+2.654\cos(x_1)+0.334\cos(3.5x_1)-0.437\cos(2.5x_2)} & x_1 \geq 0,\, x_2 \geq 0 \\ 0 & \text{otherwise} \end{cases}$ |

