# OpenReview forum: "MESSY Estimation: Maximum-Entropy based Stochastic and Symbolic densitY Estimation"
_TMLR — Accepted by TMLR_

### Review · Reviewer_3RF5 · 2023-08-21

**Summary Of Contributions:**

The central idea in this work is to draw iid samples from an unknown distribution $f$ and subsequently measure the distance between an ansatz $\hat f$ and $f$ by evolving these samples using an SDE where the drift is the gradient of the log likelihood of $\hat f$. Because an ensemble of particles evolving under this SDE converges weakly to the solution of an associated Fokker Planck equation, the relaxation time under this dynamics provides a measure of the quality of the estimator $\hat f$. By introducing basis vectors and solving the linear problem when the relaxation rate is set to its limiting value (zero), the authors obtain a solution for $\hat f$ without optimization. This procedure is subsequently used for symbolic regression. Numerical experiments on a few 1d examples are conducted and largely illustrate that MESSY obtains better solutions than convention KDE.

**Audience:**

Yes

**Broader Impact Concerns:**

None.

**Claims And Evidence:**

No

**Requested Changes:**

The related methods section should discuss Stein Variational gradient descent with a particular emphasis on moment matching variants of SVGD.

Some discussion of sample problems for which SR would be natural would help motivate the objective of the paper.

The claims in Sec. 4.1 should be clearly justified. For example, I highly skeptical of the claim that the proposed method avoids the curse of dimensionality in any meaningful sense. This statement requires a clear argument, such as a convergence rate bound.

Finally, I do not think the numerical experiments are sufficient to support the claims made in the paper. It is imperative that the authors consider a high-dimensional example. Furthermore, direct comparison to more modern methods (e.g., SVGD) should be made. If I understand correctly, the symbolic regression task could be implemented once any estimator is obtained.

**Strengths And Weaknesses:**

Strengths:

The manuscript elegantly introduces the idea of using gradient flow relaxation rates as an objective for optimizing a representation of the target density. In large part, the exposition is clear and not overly technical, while still mostly precise. Extending the density estimation problem to symbolic regression specialize the scope of the method, which in turn perhaps narrows the number of methods to which it should be compared.

Weaknesses:

Connections with the existing literature on density estimation, especially variational methods using the kernelized Stein discrepancy, are not really sufficient to determine the novelty of this approach beyond the symbolic regression problem.

Some of the claims are, I think, overstated or made without a clear argument. Especially in Sec. 4.1.

When are the relaxed existence requirements relevant?

What does it mean to avoid the curse of dimensionality in this setting?

Finally, I think the appropriate set of problems for this method could be more clearly highlighted. Much of the current ML literature on density estimation uses either nonparameteric density estimation to obtain convergence bounds or parametric estimators based on neural networks. If neither of these approaches are appropriate for the class of tasks that MESSY is intended to apply to, that could also be highlighted more cleary.

---

> ### Author Response · Authors · 2023-10-25
> **Response to Reviewer 3RF5**
>
> We thank the reviewer for their time and valuable comments.
>
> ### **Weaknesses**
> > Connections with the existing literature on density estimation, especially variational methods using the kernelized Stein discrepancy, are not really sufficient to determine the novelty of this approach beyond the symbolic regression problem.
>
> We thank the referee for bringing our attention to Stein Variational Gradient Descent method and its moment matching variation. We added an explanation in the introduction of the paper. Although the SVGD method is an attractive approach for approximating a target density by  evolving samples/particles using a  dynamical system, it is different from the work presented here in one important aspect. Our objective in the present paper is to estimate the probability density function given samples/particles rather than produce samples from this density.
>
> > Some of the claims are, I think, overstated or made without a clear argument. Especially in Sec. 4.1.
>
>  We have revised the claims made in Sec. 4.1 and added further explanations.
>
> > When are the relaxed existence requirements relevant?
>
> The relaxed existence requirement is relevant when the target moments are in the limit of realizability (target realizable moments are in the neighborhood of unrealizable moments). The standard MED optimization problem fails in this limit, since the line search in Newton-Raphson method loses accuracy given high condition number of the Hessian matrix; see for example [Alldredge, Graham W., et al., J. Comput. Phys., 258 (2014): 489-508.].
>
> The proposed non-iterative solution to the MED problem based on Gradient flow avoids this problem, as it orthogonilizes basis functions once and does not need any iterations to find Lagrange multipliers.
>
> > What does it mean to avoid the curse of dimensionality in this setting?
>
> In this paper, when we refer to curse of dimensionality, we refer to curse of dimensionality in integration. This has been clarified in the manuscript.
>
> For the standard nonlinear optimization problem of finding the MED, the Lagrange multipliers $\boldsymbol\lambda$ of the MED, i.e. $\hat f(\boldsymbol{x}) = \exp(\boldsymbol\lambda \cdot \boldsymbol H(\boldsymbol x) )/Z$, are often computed using the Newton-Raphson method. Since the initial guess for $\boldsymbol\lambda$, and hence $\hat f$, can be far from the target MED solution, the given samples cannot be used to compute the gradient and Hessian of the objective function, see eq.~A.3-4 in the Appendix A. Therefore, one either has to sample the guessed distribution and compute these moments with additional cost, or use deterministic integration techniques. In practice, deterministic numerical integration schemes are used which suffer from the curse of high dimensionality, i.e. $\mathrm{cost} \sim \mathcal{O}(N^{\mathrm{d}} )$ where $N$ is the number of discritization points in each dimension and $d$ is the dimension of $\boldsymbol x$.
>
> In the proposed linear solution, the Lagrange multipliers are computed directly using the given samples without any use of deterministic integration method. Therefore, the cost of evaluating the integrals remains linear with number of samples $N$, i.e. $\mathcal{O}(N)$. We added further clarification in the manuscript in the section 4.1.
>
> > Finally, I think the appropriate set of problems for this method could be more clearly highlighted. Much of the current ML literature on density estimation uses either nonparameteric density estimation to obtain convergence bounds or parametric estimators based on neural networks. If neither of these approaches are appropriate for the class of tasks that MESSY is intended to apply to, that could also be highlighted more cleary.
>
> We thank the reviewer for their comment. We added further explanation in the conclusion of the paper to motivate the use of MESSY for applications like recovering dynamical system given samples, etc.
>
> In this paper, we tried to pave the way for efficient least biased density estimation, i.e. MED, along with introducing further flexibility by deploying symbolic regression. We believe this density estimator may have a large impact on ML training of statistical data, as the  parameters of density fit to samples are computed in a tractable fashion.

---

> > ### Author Response · Authors · 2023-10-25
> > **Continuation of Response to Reviewer 3RF5**
> >
> > ### **Requested Changes**
> > > The related methods section should discuss Stein Variational gradient descent with a particular emphasis on moment matching variants of SVGD.
> >
> > Added.
> >
> > > Some discussion of sample problems for which SR would be natural would help motivate the objective of the paper.
> >
> > Added.
> >
> > > The claims in Sec. 4.1 should be clearly justified. For example, I highly skeptical of the claim that the proposed method avoids the curse of dimensionality in any meaningful sense. This statement requires a clear argument, such as a convergence rate bound.
> >
> > As we explained above, we are referring to the curse of dimensionality in integration. We have revised and clarified this section of the manuscript and reduced the claims. Also, we reported further results to back the claims.
> >
> > > Finally, I do not think the numerical experiments are sufficient to support the claims made in the paper. It is imperative that the authors consider a high-dimensional example. Furthermore, direct comparison to more modern methods (e.g., SVGD) should be made. If I understand correctly, the symbolic regression task could be implemented once any estimator is obtained.
> >
> > We point out that this paper is focused on the problem of finding the probability density function given samples, rather than a resampling  method. The two main claims/takeaways of the paper are:
> > - The Lagrange multipliers of MED can be computed by simply solving a linear system of equations.
> > - Accurate density recovery does not necessarily require higher-order moments. In fact, increasing the number of complex but low-order symbolic basis functions leads to better conditioning and enhanced expressiveness.
> >
> > Our experiments are carefully designed to target these two main claims. In the revised manuscript, we included examples from higher dimensional problems.
> > Our understanding is that SVGD provides discrete (particle/sample) approximation of the target density. However, here we are interested in finding a continuous, parameterized and closed form approximation.

---

### Review · Reviewer_h7mG · 2023-09-06

**Summary Of Contributions:**

The authors propose a method for density estimation based on a combination of Langevin dynamics, maximum-entropy principle, moment matching and symbolic regression. They first define an exponential family ansatz motivated by entropy maximization. They then show how to estimate its parameters by computing relaxation rates. In the second part, they propose a Monte Carlo symbolic regression method to find closed-form moments beyond the usual monomials which result in a better conditioned linear systems for the related Lagrange multipliers. They apply the resulting method to simple bimodal univarate distributions and report good results.

**Audience:**

Yes

**Broader Impact Concerns:**

--

**Claims And Evidence:**

No

**Requested Changes:**

All of the following are critical in my opinion:

- better organization of the manuscript, in particular with regard to positioning in the literature and a precise statement of the adopted setting, assumptions, notation, metrics, etc, early on in the manuscript
- a better comparison with state of the art density estimation
- comparisons with KDE with optimal data-driven kernel bandwidth choice
- a comparison with a histogram estimate with an optimal bin size
- (very critical) experiments on high-dimensional data
- ablations to test the importance of the different design choices
- more details about the used symbolic regression and the rationale for the related parameter choices
- many more ground truth densities in various dimensions, including those coming from applications, and including real rather than synthetic datasets

**Strengths And Weaknesses:**

## Strengths

- Despite the great amount of activity around symbolic regression in scientific machine learning over the last ten or so years, there are very few studies that address symbolic regression of probability density functions from finite samples, arguably a more difficult problem than "simply" fitting time series data. Given the central importance of density estimation in machine learning such efforts are clearly commendable.

- Relying on the maximum entropy principle in density estimation, especially with explicit symbolic expressions, makes good sense. It is also a nice idea to look for moments which result in well-conditioned linear system for the Lagrange multipliers. This strategy results in an effective model in the reported computer experiments on simple bimodal univariate distributions.

- In general, it is exciting to think about how symbolic regression could be used to facilitate the usually complicated problems of connecting score with density and in general work with the structured distributions we encounter in scientific ML.

## Weaknesses
I feel that the organization and motivation of the manuscript could be improved. More importantly, the computer experiments are lacking in many aspects.

- Definition 5.1 is not directly used in the paper. Why not start directly from Definition 5.2? Perhaps there is actually no need to state these problems as formal mathematical definitions? Moreover, Definition 5.1 defines supervised learning, not symbolic regression, unless it says much more about how \mathcal{S} is structured.

- With the exception of Table I in the appendix there is no discussion about the used space of symbolic expressions. From Table I it seems that all expressions consist of compositions and products of sin, cos, and monomials. It is not clear what the advantage of generating symbolic expression at random is as opposed to, say, random Fourier features or random neural networks. The main reason to use things other than monomials appears to be the improved conditioning of the linear system for the multipliers and the expressions listed in Table I are hardly interpretable. Since symbolic regression is central to the proposed method this lack of information is strange.

- **(Major) Mismatch Between Claims and Evidence**: The manuscript emphasizes its relevance in high-dimensional contexts. Quoting from page 5, "With applications to high-dimensional problems in mind, instead of looking for solutions of Eq. 2, we choose to work with appropriate empirical moemnts of this equation, ..." Yet the computer experiments focus solely on very simple one-dimensional scenarios. To make a strong case for high-dimensional applicability, it is crucial to provide experiments or discussions specific to these contexts. I imagine that this may work in dimensions 2 or 3 (although already with difficulties; see similar issues in symbolic regression methods like SINDy), but it would face serious issues going beyond that, let alone to what is usually considered high-dimensional in modern machine learning.

- There are additional major concerns with computer experiments. Although the authors deal with various operators (e.g., +, −, ×, ÷) and unary functions (e.g., cos, sin, exp, log), the computer experiments only estimate a simple bimodal distribution, with a very simple (quadratic or polynomial) exponent.

- Can the authors compare their results directly with histogram density? In Figure 2(c), I wonder how the histogram density looks with 10,000 samples. I suspect it may not be very far from the true pdf.

- On a related note: I have some doubts about the comparison with KDE. It seems that Silverman's kernel bandwidth systematically overestimates the optimal bandwidth. How about using an oracle estimate for the bandwidth? Or a cross-validated / bootstrap estimate from the data? I believe that it would result in an excellent fit. Given how complex the proposed method is, a small search for the optimal bandwidth seems trivial.

- The work should be properly situated in the literature and the setting should be made precise and explicit early on. Since you only work in 1D and only sample formulas randomly, the relevance for contemporary machine learning is not evident.

- There is a huge amount of work on density estimation, especially in the high-dimensional setting, which seems highly relevant to this manuscript yet is not mentioned. An example that comes to mind is the recent work of Stèphane Mallat and collaborators on multiscale approaches inspired by the renormalization group. These in particular seem related to the multilevel scheme proposed in the manuscript, though rigorously motivated.

- Many ideas in the first part of the paper come from Liu 2017. It is not clear for example why the proof of Proposition 1 is reproduced (but not in full). At any rate, one should acknowledge in the statement of the proposition or before that this is taken from earlier work.

## Minor

- I find the terminology "moment conservation" a bit strange in this context. I am used to seeing "moment matching" / "method of moments" (e.g., https://en.wikipedia.org/wiki/Method_of_moments_(statistics) and http://proceedings.mlr.press/v37/li15.pdf)


- Please also specify what you mean by a "least biased distribution". I understand the relation to maximum entropy but if you are really referring to bias that should be properly defined.
- After equation 4: \bf{x}-> \infty should be $\| \bf{x} \|$
- In the appendix, H should be \bf{H}?
- In the paragraph before equation (10), should the entropy be minimized or maximized?

---

> ### Author Response · Authors · 2023-10-25
> **Response to Reviewer h7mG**
>
> We thank the reviewer for recognizing the novelty and contribution presented in the paper, and for providing valuable comments which helped us improve the quality of the manuscript.
>
> ## **Weaknesses**
>
> > Definition 5.1 is not directly used in the paper. Why not start directly from Definition 5.2? Perhaps there is actually no need to state these problems as formal mathematical definitions? Moreover, Definition 5.1 defines supervised learning, not symbolic regression, unless it says much more about how $\mathcal{S}$ is structured.
>
> We thank the reviewer for pointing this out. We agree that Definition 5.1 is defining supervised learning rather than SR, and hence we have updated the definition in the manuscript. Note that Definition 5.1 serves as a brief preliminary for our main problem, which would be useful for those who are not familiar with Symbolic Regression (SR). Also, we write them as definitions to highlight the difference between the traditional SR task and our proposed task of recovering pdfs using SR.
>
> >With the exception of Table I in the appendix there is no discussion about the used space of symbolic expressions. From Table I it seems that all expressions consist of compositions and products of sin, cos, and monomials. It is not clear what the advantage of generating symbolic expression at random is as opposed to, say, random Fourier features or random neural networks. The main reason to use things other than monomials appears to be the improved conditioning of the linear system for the multipliers and the expressions listed in Table I are hardly interpretable. Since symbolic regression is central to the proposed method this lack of information is strange.
>
> We agree with the referee that the space of functions created with composition/product of sin, cos and monomials is limited. Clearly, this space can be extended by incorporating other classes of functions. Regardless of the choice for the deployed function generator, the task of symbolic regression here is to find a vector of functions that best represent data for the Maximum Entropy distribution function, i.e. leading to a well-conditioned inverse problem. In principle, random Fourier modes or random neural network may be incorporated in the random search for appropriate basis functions $\textbf H$ during the re-population step.
>
>
> We would like to point out that the interpretability of the resulting symbolic expression is dependent on the application. For example, in particle systems, the choice of $x^2$ as one of the basis functions $\textbf H$ is justified as its moment has a physical meaning, i.e. related to the energy of the system. Yet, the relation between physical moments of interest and other important but not trivial data-driven moments can be obtained through such symbolic expression for the density. Also, in the case of learning the governing evolution equation for the pdf, being able to find the distribution functions in symbolic form can help us in learning the underlying stochastic process from a given sequence of samples. For example, one can learn symbolically the transition probability corresponding to samples that follow an unknown Fokker-Planck equation (which is in exponential form), and discover the underlying dynamics accordingly. These points are now further discussed at the end of the conclusion section.

---

> ### Author Response · Authors · 2023-10-25
> **Continuation of Response to Reviewer h7mG**
>
> >(Major) Mismatch Between Claims and Evidence: The manuscript emphasizes its relevance in high-dimensional contexts. Quoting from page 5, "With applications to high-dimensional problems in mind, instead of looking for solutions of Eq. 2, we choose to work with appropriate empirical moments of this equation, ..." Yet the computer experiments focus solely on very simple one-dimensional scenarios. To make a strong case for high-dimensional applicability, it is crucial to provide experiments or discussions specific to these contexts. I imagine that this may work in dimensions 2 or 3 (although already with difficulties; see similar issues in symbolic regression methods like SINDy), but it would face serious issues going beyond that, let alone to what is usually considered high-dimensional in modern machine learning.
>
> The curse of dimensionality is avoided in the computation of integrals (gradient and Hessian) that are needed in estimating Lagrange multipliers, since in the proposed solution we compute $d$-dimensional integrals using samples. This is in contrast to the standard optimization problem of MED where samples of the initial guess is not available requiring the use of quadrature methods for the computation of the gradient and Hessian in each iteration of Newton's method. This further discussed in the section 4.1.
>
>
> We added two examples of 2-dimensional distribution function recovery using the MESSY approach. We further report on the cost of the proposed method for recovering a $d$-dimensional multivariate normal distribution. We consider two cases:  the execution time for finding MED against dimension $d=1,...,10$, for a fixed number of samples;  second, we measure the execution time against number of samples for a fixed dimension $d$. Our results show that while the integration cost of the proposed solution remains linear with number of samples for a fixed dimension, the cost  associated with solving the linear system for Lagrange multipliers increases cubically with number of bases. We note that the number of bases grows with dimension and the exact rate depends on how d-dimensional bases are created.
>
>
> Regarding the cost of SINDy and PDE-FIND with respect to dimension, we would like to point out the following. Since in these methods the regression is performed on discrete points of space, a large number of variables per dimension is needed which makes the optimization task extremely expensive even for 2D or 3D problems. Often, proper orthogonal decomposition (POD) is used to lower the dimensionality of such representation, yet it is not clear if it can be used for non-linear problems. The benefit of finding the parameterized estimate of the density is in fact in such applications. With a few basis functions and corresponding Lagrange multipliers, the joint MED distribution can represent the data with the least bias among all possible distributions. Such a continuous and differentiable representation may allow us to tackle the task of finding the governing law from data without the need for discretizing the space.
>
> >There are additional major concerns with computer experiments. Although the authors deal with various operators (e.g., $+, -, \times, \div$) and unary functions (e.g., cos, sin, exp, log), the computer experiments only estimate a simple bimodal distribution, with a very simple (quadratic or polynomial) exponent.
>
> We would like to point out that the considered test cases are the most challenging for the  standard MED method. For example, the bi-modal distribution in the limit of realizability is an extremely difficult test case for the standard MED method as the condition number of Hessian becomes large and the Newton-Raphson method fails, see [Alldredge, Graham W., et al., J. Comput. Phys., 258 (2014): 489-508.] for example.
>
> >Can the authors compare their results directly with histogram density? In Figure 2(c), I wonder how the histogram density looks with 10,000 samples. I suspect it may not be very far from the true pdf.
>
> We have added comparison with a histogram  in the appendix of the revised manuscript. Please note that the target of this work is to find/recover a continuous and differentiable estimate for the underlying probability distribution function with the least bias. Although the histogram gives reasonably accurate estimate in the limit of large number of samples, it does not provide us with a parametric, continuous and differentiable description, which can be used for interpreting and analysing the underlying pdf.

---

> > ### Author Response · Authors · 2023-10-25
> > **Continuation of Response to Reviewer h7mG**
> >
> > >On a related note: I have some doubts about the comparison with KDE. It seems that Silverman's kernel bandwidth systematically overestimates the optimal bandwidth. How about using an oracle estimate for the bandwidth? Or a cross-validated / bootstrap estimate from the data? I believe that it would result in an excellent fit. Given how complex the proposed method is, a small search for the optimal bandwidth seems trivial.
> >
> > We would like to thank the referee for pointing this out. We have tried to use an oracle estimate as well as cross-validation for finding the optimal bandwidth. It turned out that cross-validation leads to the best estimator in our examples. So we have reproduced all our experiments for KDE with a cross-validated bandwidth instead of Silverman's rule. We have modified the manuscript accordingly.
> >
> >
> > We also appreciate the referee's comment about using a more advanced version of KDE. Please note that we show KDE estimate just as a reference solution for the reader. The goal of this work is to show how one can find the least-biased parametric distribution function given samples, and not making the claim that it is more accurate than non-parametric estimators. Please note that non-parametric estimators may converge to the true solution as number of samples increases, while parametric estimators may converge as number of parameters increases. Furthermore, given that KDE is non-parametric and not interpretable, we do not pursue a strict comparison here.
> >
> > >The work should be properly situated in the literature and the setting should be made precise and explicit early on. Since you only work in 1D and only sample formulas randomly, the relevance for contemporary machine learning is not evident.
> >
> > The two main points introduced in the paper are: \
> > i) MED Lagrange multipliers can be simply solved using a linear system of equations.\
> > ii) Better density recovery does not necessarily require matching higher-order moments; in fact, matching a higher number of complex (low-order) symbolic basis functions leads to better conditioning and density recovery.
> >
> > The experiments we show are carefully designed to demonstrate these two points. In the revised manuscript we added 2D examples and discuss how the computational cost increases with the dimension $d=1,...,10$.
> >
> > >There is a huge amount of work on density estimation, especially in the high-dimensional setting, which seems highly relevant to this manuscript yet is not mentioned. An example that comes to mind is the recent work of Stèphane Mallat and collaborators on multiscale approaches inspired by the renormalization group. These in particular seem related to the multilevel scheme proposed in the manuscript, though rigorously motivated.
> >
> > We thank the referee for bringing our attention to the interesting development on wavelet method. We have added references to wavelet method and recent work of Stèphane Mallat in the introduction.
> >
> >
> > We would like to point out that in the wavelet method, the domain is decomposed into a multi-grid/scale and on each level the coefficients of the orthogonal wavelet basis functions are computed. These bases are used for estimating the conditional probability between the levels and it is not clear how much bias is introduced. Having said that, it seems an interesting approach and we are considering combining MED with wavelet method in our future work.
> >
> >
> > Furthermore, we point out that the multi-level method suggested in our paper is different from the multi-grid/scale domain decomposition in the wavelet method. Here, at each level, first we  remove a portion of samples that are recovered by MED in the previous level, and then find the fit for the remaining samples. Hence, there's no domain decomposition, at least in the form suggested in the  wavelet method.
> >
> > >Many ideas in the first part of the paper come from Liu 2017. It is not clear for example why the proof of Proposition 1 is reproduced (but not in full). At any rate, one should acknowledge in the statement of the proposition or before that this is taken from earlier work.
> >
> > The proof of proposition 1 is reiterated here to help the reader understand why the stationary solution of the proposed Fokker-Planck equation is $\hat{f}$ without getting too technical. We made a reference to Liu 2017 at the end of the proof for more details.

---

> ### Author Response · Authors · 2023-10-25
> **Continuation of Response to Reviewer h7mG**
>
> ## **Minor**
> >I find the terminology "moment conservation" a bit strange in this context. I am used to seeing "moment matching" / "method of moments" (e.g., https://en.wikipedia.org/wiki/Method_of_moments_(statistics) and http://proceedings.mlr.press/v37/li15.pdf)
>
> Revised.
>
> >Please also specify what you mean by a "least biased distribution". I understand the relation to maximum entropy but if you are really referring to bias that should be properly defined.
>
> By bias, we are referring to error in other constraints/information that are not used in finding the distribution function, e.g. error in moments other than those that are matched. In information theory, it is found that among all distributions that match given constraints,  the one that minimizes the Shannon entropy (or maximizes it with the sign difference) is the least-biased (or most likely) distribution. We refer the referee to [Khinchin, Aleksandr Iakovlevich. Mathematical foundations of information theory, Vol. 434. Courier Corporation (1957).] and  [Kapur, Jagat Narain. Maximum-entropy models in science and engineering. John Wiley \& Sons, 1989.] for more details.
>
> >After equation 4: $\bf{x}\rightarrow \infty$ should be $|\bf x|$.
>
> Corrected.
>
> >In the appendix, H should be $\bf{H}$.
>
> Corrected.
>
> >In the paragraph before equation (10), should the entropy be minimized or maximized?
>
> The MED either minimizes the Shannon entropy $\int \mathcal{F} \log(\mathcal{F}) d \textbf x$ which is a convex functional, or maximizes the physical entropy $-\int \mathcal{F} \log(\mathcal{F}) d \textbf x$ which is a concave functional. In this paper we used the former definition, therefore the text is correct.
>
> ## **Requested Changes:**
> >All of the following are critical in my opinion:
>
> >better organization of the manuscript, in particular with regard to positioning in the literature and a precise statement of the adopted setting, assumptions, notation, metrics, etc, early on in the manuscript
>
> We have made some changes and revised the manuscript to address the referee's concerns.
>
> >a better comparison with state of the art density estimation
>
> We would like to point out that Maximum Entropy Distribution is the least biased parametric density estimator and this work proposes a simple solution to finding Lagrange multipliers with a cost that is linear with number of samples.
>
> >comparisons with KDE with optimal data-driven kernel bandwidth choice
>
> As we mentioned above, we have reproduced KDE results with a cross-validated optimal bandwidth. We have updated the manuscript to reflect this change. The KDE estimator is shown as a reference for the reader. KDE is non-paramteric and not interpretable, hence it does not relevant to compare MESSY with the most accurate version of KDE.
>
> >a comparison with a histogram estimate with an optimal bin size
>
> As we mentioned above, we have added  comparison with histogram estimate in the appendix of the revised manuscript. Please note that  histogram is not a parametric, continuous, differentiable, and interpretable density estimator. Therefore, similar to KDE, its comparison is shown only as a reference for the reader.
>
> >(very critical) experiments on high-dimensional data
>
> As we mentioned above, we have added 2D examples  in the revised manuscript. We also report how the cost of MESSY-P scales with dimension $d$ for $d=1,...,10$.
>
> >ablations to test the importance of the different design choices
>
> We performed an ablation study to show the effect of each component in the proposed MESSY solution to the MED estimate. The four components of MESSY Estimation are: i) the symbolic component, ii) the multi-level component, iii) the orthogonalization, and iv) the cross-entropy step. Results are added to the appendix.
>
> >more details about the used symbolic regression and the rationale for the related parameter choices
>
> We added more details to the symbolic regression task that we deployed in the paper. Please note that this work is a proof of concept. Clearly more complicated symbolic regression method can be used to optimized the proposed solution.
>
> >many more ground truth densities in various dimensions, including those coming from applications, and including real rather than synthetic datasets.
>
> In this paper we present a simple solution to the MED estimate and show the possibility of using symbolic regression to deal with its ill-conditionality. Here, we showed several 1D and 2D test cases, and analysied the cost with respect to dimension $d=1,...,10$ and number of samples in estimating $d$-dimensional distribution function. The  examples shown here are meant as a proof of concept. In future work, we intend to test the proposed estimator on higher dimensional problem with raw data and more focus on its applications in Machine Learning.

---

### Review · Reviewer_tPUb · 2023-10-13

**Summary Of Contributions:**

The paper proposes a novel algorithm for density estimation. The approach considers the evolution of the moments of certain basis functions when a density follows a diffusion process (overdamped Langevin dynamics) having the estimated density function as its stationary distribution. This rate of evolution is used to construct an objective that allows estimation of the Lagrange multipliers corresponding to the maximum entropy distribution w.r.t a set of basis functions. The estimated multipliers depend on certain expectations w.r.t the empirical measure of samples from the true density. The objective relies on the observation that the rate of evolution of the expectations w.r.t the dynamics at the true density is $0$ when the estimated density matches the true density. The approach further utilizes symbolic regression to learn suitable basis functions. The paper discusses computational aspects of the proposed approach and verifies its effectiveness on certain one-dimensional densities.

**Audience:**

Yes

**Broader Impact Concerns:**

The paper is primarily of a theoretical nature with simple toy experiments. Therefore, there are no concerns regarding its broader impact.

**Claims And Evidence:**

Yes

**Requested Changes:**

Adjustments critical for securing recommendation for acceptance:

- It should be clarified that Eq.1 is the overdampled Langevin dynamics corresponding to the density $\hat{f}$ and differs from the score-based generative model in Song et al. (2020).
- The optimization problem corresponding to $g(t)$ should be clearly defined and motivated.

Adjustments that will strengthen the work:

- Experiments on multi-dimensional distributions.
- Discussion of sample and time complexity bounds for the proposed and existing approaches for density estimation.

**Strengths And Weaknesses:**

Strengths:

- The proposed approach appears to be significantly novel.
- The paper provides justification for different components of the approach. For instance, to illustrate the benefit of symbolic regression, a comparision is made between MESSY-P and MESSY-S, algorithms utilizing a polynomial basis and a basis obtained using symbolic regression respectively.
- The paper discusses different practical aspects of the approach such as the computational cost and conditioning of optimization problems.
- The non-technical sections of the paper are well-written.

Weaknesses:

- Some terms in the paper are not clearly defined. For instance, in the line above equation 8 "Here we note that the Hessian for this optimization", the "optimization" problem isn't clearly defined or explained. Only after Proposition 4.3 and Eq. 16, it becomes apparent that "g(t)" is being interpreted as a gradient for an associated optimization problem on the Lagrange multipliers. This should be explained clearly with precise definitions before statements such as "These relaxation rates can be used as the gradient in the search for parameters of a given ansatz" above eq. 7.
- Certain references to the literature are innacurate. The paper refers to Eq.1 as the "score-based generative model" in Song et al. (2020). However in the reverse-SDE utilized in Song et al. (2020), the "score" is the grad-log of the density at the given time t, not that of a fixed density $\hat{f}$ corresponding to the stationary distribution. Instead, Eq.1 should be more appropriately referred to as the overdampled Langevin dynamics for $\hat{f}$ .
- The approach lacks bounds on sample complexity (with a fixed set of basis functions). In contrast, the sample rates for existing approaches such as Kernel density estimation (KDE) have been extensively analyzed.
- The approach is evaluated only on a few simple one-dimensional distributions. This makes it difficult to assess the significance of the improvements.

---

> ### Author Response · Authors · 2023-10-25
> **Response to Reviewer tPUb**
>
> We thank the reviewer for the positive feedback and helpful comments.
>
> ### **Weaknesses**
> > Some terms in the paper are not clearly defined. For instance, in the line above equation 8 "Here we note that the Hessian for this optimization", the "optimization" problem isn't clearly defined or explained. Only after Proposition 4.3 and Eq. 16, it becomes apparent that "g(t)" is being interpreted as a gradient for an associated optimization problem on the Lagrange multipliers. This should be explained clearly with precise definitions before statements such as "These relaxation rates can be used as the gradient in the search for parameters of a given ansatz" above eq. 7.
>
> We thank the reviewer for pointing this out. We have revised the manuscript to explicitly describe the proposed optimization problem for the new approach of finding the Lagrange multipliers of MED estimate.
>
> > Certain references to the literature are innacurate. The paper refers to Eq.1 as the "score-based generative model" in Song et al. (2020). However in the reverse-SDE utilized in Song et al. (2020), the "score" is the grad-log of the density at the given time t, not that of a fixed density $\hat{f}$ corresponding to the stationary distribution. Instead, Eq.1 should be more appropriately referred to as the overdampled Langevin dynamics for $\hat{f}$.
>
> We have have revised the manuscript accordingly.
>
> > The approach lacks bounds on sample complexity (with a fixed set of basis functions). In contrast, the sample rates for existing approaches such as Kernel density estimation (KDE) have been extensively analyzed.
>
> For a fixed number of basis functions, the only source of error is the Monte Carlo integration (Law of Large Numbers) where the variance of estimate is $\mathcal{O}(1/N)$ leading to  the root of mean squared error $\mathcal{O}\big(1/\sqrt{N}\big)$ for $N$ samples. We added further explanation in section 4.1, where we discuss how the method avoids the curse of dimensionality in integration.
>
> > The approach is evaluated only on a few simple one-dimensional distributions. This makes it difficult to assess the significance of the improvements.
>
> The two key ideas presented in the paper are:
> - The Lagrange multipliers of MED can be efficiently found by simply solving a linear system of equations.
> - Better density recovery does not necessarily require higher-order moments; increasing the number of complex (low-order) symbolic basis functions leads to better conditioning and enhanced expressiveness.
>
> Our experiments are carefully designed to demonstrate these two ideas. We added 2-dimensional examples to the manuscript and a discussion, as well as results on how the cost increases with dimension. A promising future direction would be testing the proposed method on high dimensional distributions from raw data.
>
> ### **Requested Changes**
> > It should be clarified that Eq.1 is the overdampled Langevin dynamics corresponding to the density $\hat{f}$ and differs from the score-based generative model in Song et al. (2020).
>
> Revised accordingly.
>
> > The optimization problem corresponding to $g(t)$ should be clearly defined and motivated.
>
> We added definition of the optimization problem for standard and proposed method of finding Lagrange multipliers in definition 4.3 and 4.4, respectively.
>
> > Experiments on multi-dimensional distributions.
>
> As we mentioned above, we have added 2-dimensional examples in the revised manuscript. Also, we discussed the cost of the proposed MESSY-P in recovering a $d$-dimensional distribution function where $d=1,...,10$.
>
> > Discussion of sample and time complexity bounds for the proposed and existing approaches for density estimation.
>
> Added a discussion in section 4.1.

---

### Decision · Action_Editor_RFhq · 2024-01-17

**Recommendation:** Accept with minor revision

**Comment:**

The paper presents an algorithm for density estimation using overdamped Langevin dynamics and symbolic regression. This method, which focuses on the evolution of moments to determine Lagrange multipliers for maximum entropy distribution, assesses accuracy based on the alignment with true density dynamics. While the algorithm shows accuracy and introduces new ideas, there is uncertainty about its effectiveness in high-dimensional data, as the current research is limited to simple one-dimensional cases despite claims of potential high-dimensional applicability.

**Audience:**

Given TMLR's criteria, the paper's algorithm for density estimation and its implications in machine learning are likely to interest TMLR's audience. Howver, by addressing the additional points and possible limitations on high-dimensional data , the paper will be offering a better-rounded perspective on its applicability and future research directions. This will enrich the paper's value to TMLR's audience and align with the journal's standards for technical depth and practical relevance.

**Claims And Evidence:**

The paper introduces a density estimation algorithm that leverages the evolution of moments in overdamped Langevin dynamics to derive Lagrange multipliers for maximum entropy distribution. It uses the alignment of these moments' evolution with the true density dynamics as a key indicator of accuracy. The study also employs symbolic regression for basis function selection and validates its approach with low-dimensional density experiments.

Despite acknowledging the method's accuracy and its introduction of innovative concepts, reviewers however collectively note its ambiguous applicability to high-dimensional data. The manuscript itself still underscores its possible relevance in high-dimensional scenarios, as stated on page 5: "With applications to high-dimensional problems in mind, instead of looking for solutions of Eq. 2, we choose to work with appropriate empirical moments of this equation, ..." However, the empirical studies presented are limited to simplistic one-dimensional cases.

Based on the review and evaluation of the manuscript, I thus recommend acceptance of the paper. However, it is crucial for the author to explicitly address certain limitations in the study and to expand the discussion in the conclusion, particularly regarding the significance of testing these ideas in higher dimensions First, the author should clearly delineate the limitations of the current approach. Secondly, the conclusion section needs to be expanded to emphasize the importance of extending this research into higher-dimensional spaces. Given the complexity and relevance of high-dimensional data in modern machine learning, it's imperative to discuss how the proposed algorithm might (or not)perform or be adapted to these contexts.

---

> ### Author Response · Authors · 2024-01-31
> **Minor Points Addressed**
>
> We thank the Action Editor for reviewing our work. \
> Motivated by the reviewer comments, in the revised manuscript we tested the proposed method on high dimensional problems and showed how the cost increases with dimension (up to a 10-dimensional sample space). That said, we agree with the Action Editor that more studies need to be carried out in future works for targeted Machine Learning tasks. Hence, we have carefully revised the manuscript to temper the claims made (particularly in relation to the applications to high-dimensional problems).
>
> As per your suggestion, we have made the following modifications in the camera-ready version:
>
> - We have re-written the paragraph on page 5.\
> Specifically, instead of saying: \
> "*With applications to high-dimensional problems in mind, instead of looking for solutions of Eq. (2), we choose to work with appropriate empirical moments of this equation, ...*", \
> we now simply say:\
> "*Instead of seeking solutions of Eq. (2), our approach focuses on working with appropriate empirical moments of this equation, ...*"
>
> - We have expanded the conclusion section to delineate the limitations of the proposed approach and to emphasize the importance of extending this work to higher-dimensional spaces:
>   - We added a paragraph discussing the limitations of this work. In particular, we added the following:\
> "*Although the proposed MESSY estimate alleviates a number of  numerical challenges with finding the MED given samples, it cannot expand the space of the MED solution. In other words, if the moment problem does not permit any MED solutions, e.g. (Junk, 1998), the MESSY estimate fails to infer any densities, since there is no MED solution to be found. Secondly, there are no guarantees that the coefficient of the leading term in the exponent of the MED solution will be negative. Hence, in unbounded domains the MESSY estimate may diverge in the tails of the distribution. Both of these issues can be resolved by regularizing the MED ansatz, e.g. via incorporating a Wasserstein distance from a prior distribution as suggested by Sadr et al. (2023).*"
>   - We have added the following sentence to the end of the first paragraph of the conclusion:\
>  "*Further detailed analysis of the trade-off between cost and accuracy in inferring densities in high-dimensional probability spaces will be addressed in future work.*"
>   - We have also added the following sentence to the end of the last paragraph of the conclusion:\
> "*That said, exploring the efficiency of our algorithm in high-dimensional contexts is a critical next step, considering the complexity of modern machine learning datasets.*"

---

> ### Author Response · Authors · 2024-01-31
> **Camera Ready Version Submitted**
>
> We would like to thank the Action Editor and the reviewers for handling our submission! We have submitted the camera-ready version.